# DPO Meets PPO: Reinforced Token Optimization for RLHF

**Han Zhong** [* 1]  **Zikang Shan** [* 1]  **Guhao Feng** [* 2]  **Wei Xiong** [3]  **Xinle Cheng** [1]  **Li Zhao** [4]  **Di He** [2]  **Jiang Bian** [4]
**Liwei Wang** [1 2]

## Abstract

In the classical Reinforcement Learning from Human Feedback (RLHF) framework, Proximal Policy Optimization (PPO) is employed to learn from sparse, sentence-level rewards—a challenging scenario in traditional deep reinforcement learning. Despite the great successes of PPO in the alignment of large language models, its open-source implementation is still largely sub-optimal. To address these issues, we introduce a framework that models RLHF problems as a Markov decision process (MDP), enabling the capture of fine-grained token-wise information. Under this framework, we introduce an algorithm Reinforced Token Optimization (RTO), which learns the token-wise reward function from preference data and performs policy optimization based on this learned token-wise reward signal. Theoretically, RTO is proven to have the capability of finding the near-optimal policy sample-efficiently. For its practical implementation, RTO innovatively integrates Direct Preference Optimization (DPO) and PPO. DPO, originally derived from sparse sentence rewards, surprisingly provides us with a token-wise characterization of response quality, which is seamlessly incorporated into our subsequent PPO training stage. Extensive experiments demonstrate that RTO performs better than PPO and other direct preference learning algorithms. In particular, RTO outperforms PPO by 7.5 points on the AlpacaEval 2 benchmark and by 4.1 points on Arena-Hard. Our code and models are available at https://github.com/zkshan2002/RTO.

[*]Equal contribution  [1]Center for Data Science, Peking University  [2]State Key Laboratory of General Artificial Intelligence, School of Intelligence Science and Technology, Peking University  [3]University of Illinois Urbana-Champaign  [4]Microsoft Research Asia.  Correspondence to:  Han Zhong <hanzhong@stu.pku.edu.cn>, Li Zhao <lizo@microsoft.com>, Di He <dihe@pku.edu.cn>, Liwei Wang <wanglw@cis.pku.edu.cn>.

*Proceedings of the 42nd International Conference on Machine Learning*, Vancouver, Canada. PMLR 267, 2025. Copyright 2025 by the author(s).

## 1. Introduction

Reinforcement Learning from Human Feedback (RLHF) has emerged as a key technique for aligning foundation models with human values and preferences (Christiano et al., 2017; Ziegler et al., 2019). It has been pivotal in enabling Large Language Models (LLMs) to produce more helpful, harmless, and honest responses (Bai et al., 2022), as demonstrated in significant applications such as ChatGPT (OpenAI, 2023), Claude (Anthropic, 2023), and Gemini (Team et al., 2023). The classical RLHF pipeline (Ziegler et al., 2019; Ouyang et al., 2022) consists of two steps: (i) Reward training from human feedback, where the learner learns the reward function based on preference data, typically through Maximum Likelihood Estimation (MLE). (ii) Reward-based RL training, where the learner employs the seminal deep RL algorithm Proximal Policy Optimization (PPO; Schulman et al., 2017) to optimize the reward learned in the previous step.

Despite the success of this framework in the aforementioned powerful closed-source LLMs, the training of PPO is known to be unstable and sample-inefficient (Choshen et al., 2019). While researchers have made efforts to propose alternative approaches to the PPO algorithm, with notable examples like rejection sampling fine-tuning (Dong et al., 2023; Gulcehre et al., 2023), direct preference learning algorithms (Rafailov et al., 2023; Zhao et al., 2023; Azar et al., 2023), there is little evidence that these newly proposed approaches alone can make the state-of-the-art LLMs. Therefore, improving the performance of the PPO algorithm in the context of RLHF is still an important research direction that is largely under-explored.

After examining the open-source implementation of PPO, we identify that one potential reason for the sub-optimal performance of PPO is the mismatch between the formulation of RLHF and the nature of PPO. Specifically, in the existing framework (Ouyang et al., 2022; Bai et al., 2022), RLHF is formulated as a *bandit*, where the entire response sentence is considered to be an action, and the reward is sentence-level, evaluating only the overall quality of the response. However, PPO is designed for multi-step RL problems modeled as *Markov decision processes* (MDPs), requiring a token-wise reward assignment to each step. In typical im-

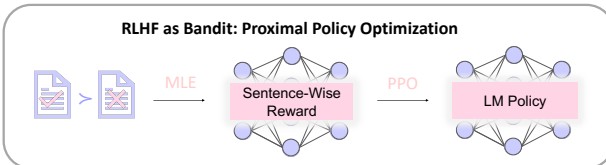 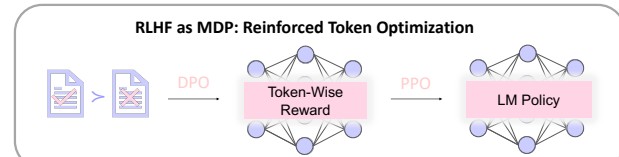

*Figure 1.* In the MDP framework of RLHF, `RTO` uses DPO to derive a token-level reward function and then applies PPO to enhance it. This approach is significantly different from the traditional RLHF process, which employs PPO to improve sentence-level rewards under the bandit framework of RLHF.

plementations of PPO (e.g., OpenRLHF and TRL), besides the regularization reward function assigned to each token to ensure the fine-tuned LLM stays close to the supervised fine-tuning (SFT) model, the learned sentence-level reward is only distributed to the last token, while other tokens receive zero learned reward. See (2.3) for the formal mathematical description. Clearly, there is a separation in terms of the assignment strategies of the regularization reward and the learned reward. Meanwhile, while it is generally believed that a fine-grained characterization with token-wise feedback can provide more information, in practice, it is also challenging to collect effective token-wise feedback for human conversations and use it in the MLE process. Consequently, the construction of token-wise reward signals also remains largely under-explored in the literature of RLHF.

**Our Contributions.** In this work, we aim to address the aforementioned issues by developing an RLHF framework with a fine-grained token-wise reward characterization, establishing the mathematical foundation, and advancing practical algorithmic designs. The key contributions of this work are summarized as follows.

- We propose a framework that models RLHF as a (KL-regularized) MDP, offering a more precise token-wise characterization of the LLM's generation process. Furthermore, we provide theoretical insights into why the token-wise MDP formulation is superior to the previous sentence-level bandit formulation of RLHF.

- Under the MDP formulation of RLHF, we introduce Reinforced Token Optimization (`RTO`), which extracts token-wise reward signals from offline preference data and subsequently performs RL training with respect to the learned token-wise rewards. Using MLE as the token-wise reward learning oracle, we prove that `RTO` can learn a near-optimal policy in a sample-efficient manner.

- Moving toward the practical implementation of `RTO`, we adopt a novel token-wise reward extraction approach from direct preference optimization (DPO; Rafailov et al., 2023). By assigning this DPO-based token-wise reward function to each token and then optimizing with PPO. `RTO` demonstrates superior performance compared to both

PPO and direct preference learning baselines such as DPO (Rafailov et al., 2023), R-DPO (Park et al., 2024), and SimPO (Meng et al., 2024). In particular, RTO achieves a 7.5-point improvement on the AlpacaEval 2 benchmark and a 4.1-point improvement on Arena-Hard. Additionally, `RTO` exhibits strong data scaling properties compared to PPO — (i) reaching PPO-level performance with only $1/8$ of the data and (ii) continuing to improve as more data is added, whereas PPO saturates early.

In summary, under the MDP formulation of RLHF, we develop a new principled RLHF algorithm, `RTO`, that leverages token-wise reward signals derived from offline preference data using DPO, and subsequently performs PPO training to optimize the token-wise rewards. The pipeline of `RTO` is visualized in Figure 1.

### 1.1. Related Works

We review the works that are mostly related to our project in this subsection. Due to the space constraint, we refer interested readers to the survey (Casper et al., 2023) for a more comprehensive overview of RLHF.

**RLHF algorithm.** As we mentioned in the introduction, tuning the PPO algorithm to its best performance requires extensive efforts and resources are often unavailable to the open-source community. Motivated by this, researchers have made efforts to develop alternative approaches to the PPO algorithm. As a direct extension of the best-of-n inference (Nakano et al., 2021), rejection sampling fine-tuning is proposed by Dong et al. (2023); Gulcehre et al. (2023); Wang et al. (2024), which prompts the LLM to generate $n$ responses per prompt and uses a learned reward function to rank the responses and fine-tune the model on those with high rewards. Besides, inspired by the reward-conditioned training in RL literature (Chen et al., 2021), Hu et al. (2023); Yang et al. (2024a) develop conditional SFT to avoid the reward learning. Another line of work aims to skip the reward modeling step and may be referred to as the direct preference learning approach (Zhao et al., 2023; Rafailov et al., 2023; Azar et al., 2023; Tang et al., 2024). Among them, the direct preference optimization (DPO) algorithm is the most popular one, mostly due to its innovative idea: *your*

*language model is secretly a reward model*. In particular, according to the reward benchmark (Lambert et al., 2024), the DPO-aligned algorithm often admits a competing ranking accuracy as a reward function. We will formally discuss the principle of DPO in Appendix C.1, which also partly motivates our methods. After these, there are also many tasks that consider the variants of this direct preference learning approach by increasing the training steps (Xiong et al., 2023; Hoang Tran, 2024) and consider the more general preference signal sources (Ye et al., 2024; Rosset et al., 2024). Although all these recently proposed algorithms achieve promising results, there is little evidence that these algorithms alone without PPO can make state-of-the-art LLMs. Therefore, understanding PPO and improving its performance in the context of foundation model alignment is still an important research direction.

**Improving PPO in the context of RLHF.** Although some works (e.g., Uesato et al., 2022; Lightman et al., 2023; Yang et al., 2024b) use token-wise or step-wise information to enhance the performance of LLMs, such as their reasoning ability, we will not discuss them in detail here. Instead, we will focus on comparing our work with others that aim to improve the PPO in RLHF. In particular, Li et al. (2023b) and Ahmadian et al. (2024) state that the PPO is not the best fit for RLHF because of the sentence-level reward and deterministic transition, and argue that the reinforce-style (Williams, 1992) algorithms perform better. Wu et al. (2024) proposes to construct several separate reward functions for different goals and use the linear combination of them to guide the PPO training, but the separate models are still confined to the sentence level. Similarly, Jang et al. (2023) extends the PPO to the multi-objective optimization scenario, but still uses the sentence-level modeling. Chan et al. (2024) shares similar insights that aim to improve PPO via a dense reward. They still follow the two-staged RLHF framework to model the reward function via MLE of the Bradley-Terry model and assume that the learned reward is based on the transformer (Vaswani et al., 2017). Then, they propose to use the attention value to redistribute the final scalar reward on a token level. In comparison, while sharing similar insights about using a token-wise reward, our techniques to obtain the dense signal and mathematical motivation are fundamentally different.

**Theoretical study of RLHF.** The theoretical study of RLHF may date back to the dueling bandit and dueling RL (e.g., Yue et al., 2012; Saha, 2021; Faury et al., 2020; Bengs et al., 2021; Pacchiano et al., 2021; Chen et al., 2022; Zhu et al., 2023; Wang et al., 2023; Zhan et al., 2023a;b), where the reward maximization problem is considered in the face of preference signals, instead of the absolute reward signals. However, the reward maximization framework admits a greedy and deterministic optimal policy, which deviates from the principle of generative AI. Meanwhile, instead of

the original reward function, the most widely used learning target is a Kullback-Leibler (KL)-regularized one. In recognition of the above issues, Xiong et al. (2023) first formally formulates the RLHF as the reverse-KL constrained contextual bandit in offline, online, and hybrid settings, and proposes sample-efficient algorithms in different settings accordingly. Beyond the reward-based framework under the Bradley-Terry model, Azar et al. (2023); Ye et al. (2024) consider the RLHF under a general preference oracle, and motivate the algorithmic design in a KL-regularized minimax game between two LLMs. In particular, Azar et al. (2023) proposes the first sample-efficient planning algorithm, and Ye et al. (2024) designs the sample-efficient learning algorithms in offline and online settings. Notably, as these studies of the KL-regularized framework align with the practical applications closely, the theoretical insights naturally motivate practically powerful algorithms like GSHF (Xiong et al., 2023), Nash-MD (Azar et al., 2023), and DNO (Rosset et al., 2024). However, we remark that Xiong et al. (2023); Azar et al. (2023); Ye et al. (2024) are still confined to the bandit setting, thus differing from the MDP formulation presented in this paper.

**Concurrent and Subsequent work.** We notice several concurrent and independent works by Rafailov et al. (2023); Zeng et al. (2024); Meng et al. (2024). Rafailov et al. (2024) also provide a token-wise MDP formulation for RLHF. Their work shares the same insight as ours, namely that "DPO implicitly optimizes the token-wise reward". Based on this insight, they improve the efficiency of search-based algorithms. In contrast, we propose a new algorithm RTO that leverages the token-wise reward functions to enhance the performance of PPO. In addition, our work provides a theoretical foundation for the unique advantages of token-wise MDP and its sample-efficient learning. Meng et al. (2024) propose SimPO by modifying the DPO objective, replacing the reference model with response length, and adding a margin threshold. Zeng et al. (2024) also consider a token-level reward and leverage this insight to develop token-level DPO, which performs better than the original DPO. These are beyond the scope of our work. In contrast, we utilize the implicit token-level reward provided by the original DPO as the dense token-level reward for RL training. We include these two algorithms as baselines to demonstrate the superior performance of RTO. Finally, following our work, Cui et al. (2025); Yin et al. (2025) utilizes implicit rewards in RL training to enhance chat and reasoning capabilities, highlighting the broad applicability of our method.

## 1.2. Notation

Given a set $\mathcal{X}$, we denote the collection of distributions over $\mathcal{X}$ by $\Delta(\mathcal{X})$. We use $\mathbb{1}\{\cdot\}$ to denote the indicator function. For any positive integer $h$, we use the notation $y_{1:h}$ to denote the sequence $\{y_1, y_2, \ldots, y_h\}$.

## 2. Preliminaries

In this section, we introduce the standard RLHF paradigm. Let $x \in \mathcal{X}$ denote the prompt sampled from a distribution $\rho \in \Delta(\mathcal{X})$, and $y = (y_1, y_2, \ldots, y_h, \ldots)$ be the corresponding response, which is a sequence of tokens generated by LLMs, where $y_i$ represents the $i$-th token. In practice, it is widely assumed (Ziegler et al., 2019; Ouyang et al., 2022) that the preference signal is generated according to the Bradley-Terry (BT) model (Bradley & Terry, 1952):

$$\mathbb{P}(y^1 \succ y^2 | x, y^1, y^2) = \sigma\big(r(x, y^1) - r(x, y^2)\big), \quad (2.1)$$

where $\sigma(z) = 1/(1 + \exp(-z))$ is the sigmoid function, and $r$ is a ground-truth reward function defined at the **sentence level**. In other words, the reward function $r$ only evaluates the overall performance of the entire response. The classical RLHF pipeline (Ziegler et al., 2019; Ouyang et al., 2022) typically consists of two steps: reward training from human feedback and reward-based RL training. In the first step, the learner is given a dataset $\mathcal{D} = \{(x, y^w, y^l)\}$, where $y^w$ denotes the preferred response over the $y^l$. The reward function is learned through Maximal Likelihood Estimation (MLE) on this dataset $\mathcal{D}$:

$$r_{\mathrm{MLE}} = \operatorname*{argmax}_r \mathbb{E}_{(x, y^w, y^l) \sim \mathcal{D}}\big[\log\big(\sigma(r(x, y^w) - r(x, y^l))\big)\big].$$
$$(2.2)$$

In the second step, the learned reward $r_{\mathrm{MLE}}$ from the previous step is optimized while ensuring that the updated language model (LLM) does not deviate significantly from the reference model $\pi_{\mathrm{ref}}$, usually selected as a supervised fine-tuned (SFT) LLM. This is because reward optimization along usually leads to reward hacking (Casper et al., 2023), meaning that the LLM will utilize the imperfection of the reward model and chase for a high reward but with a poor performance at the same time. Formally, the LLM is optimized with respect to the learned reward $r_{\mathrm{MLE}}$ with a KL-regularized term:

$$\widehat{\pi} = \operatorname*{argmax}_\pi \mathbb{E}_{x \sim \rho, y \sim \pi(\cdot | x)}\left[r_{\mathrm{MLE}}(x, y) - \beta \log \frac{\pi(y \,|\, x)}{\pi_{\mathrm{ref}}(y \,|\, x)}\right],$$

where $\beta > 0$ is an appropriate KL penalty coefficient. This KL-regularized target is widely adopted in practice (Christiano et al., 2017; Ziegler et al., 2019; Ouyang et al., 2022; Bai et al., 2022; Rafailov et al., 2023) to balance reward optimization and the goal of staying close to the reference policy. Another primary technical reason is that this regularization ensures that the framework admits a stochastic optimal policy, as compared to the deterministic greedy reward maximizer. The policy optimization step is typically achieved by PPO (Schulman et al., 2017), a seminal deep RL algorithm for solving multi-step decision-making problems and its implementation requires a reward signal at each step (corresponding to each token in the context of LLMs).

To this end, given a prompt $x$ and a response $y = y_{1:H}$ containing $H$ tokens, existing open-source implementations of PPO assign the sentence-level reward $r_{\mathrm{MLE}}(x, y)$ to the last token and optimize the following reward:

$$r_{\mathrm{ppo}}(x, y_{1:h}) \qquad\qquad\qquad (2.3)$$
$$= \begin{cases} 0 - \beta \log \frac{\pi(y_h \,|\, x, y_{1:h-1})}{\pi_{\mathrm{ref}}(y_h \,|\, x, y_{1:h-1})} & \text{if } h \leq H - 1, \\ r_{\mathrm{MLE}}(x, y) - \beta \log \frac{\pi(y_h \,|\, x, y_{1:h-1})}{\pi_{\mathrm{ref}}(y_h \,|\, x, y_{1:h-1})} & \text{if } h = H, \end{cases}$$

where $\pi$ is the current policy to be improved. However, it is well known that sparse rewards can make learning more difficult compared to dense rewards (Andrychowicz et al., 2017). One natural solution is to design dense token-wise rewards used for PPO training, but this is beyond the scope of the current bandit formulation for RLHF and motivates us to provide a framework with more fine-grained token-wise characterization that enables the use of token-wise rewards.

## 3. RLHF Formulation: From Bandit to MDP

In this section, we introduce our MDP formulation for RLHF. Section 3.1 describes how to characterize RLHF using token-wise MDPs in the context of LLMs. Section 3.2, we provide the learning objective under this framework. Lastly, Section 3.3 demonstrates the advantages of the token-wise MDP formulation compared to the sentence-wise bandit formulation.

### 3.1. MDP Formulation for RLHF

We model the RLHF problem as a Markov decision process (MDP), which is denoted as a tuple $\mathcal{M} = (\mathcal{S}, \mathcal{A}, \mathcal{P}, r, \rho, H)$. Here $\mathcal{S}$ is the state space, $\mathcal{A}$ is the action space, $\mathcal{P} : \mathcal{S} \times \mathcal{A} \to \Delta(\mathcal{S})$ is the transition kernel, $r$ denotes the reward function, $\rho$ signifies the initial state distribution and $H$ is the maximal number of interaction steps. A (Markov) policy in MDPs $\pi : \mathcal{S} \to \Delta(\mathcal{A})$ is a mapping from state to a distribution over actions. The interaction between the environment $\mathcal{M}$ and the agent can be described as follows. Initially, the starting state $s_1$ is sampled from the initial distribution $\rho$. At the $h$-th step, the agent observes the state $s_h$ and selects an action $a_h$ based on its policy. The environment then transits to the next state $s_{h+1}$, which is sampled from the distribution $\mathcal{P}(\cdot \,|\, s_h, a_h)$. This interaction continues until a certain ending condition is satisfied, which will be triggered within $H$ steps.

In the standard text generation process of large language models (LLMs), each state $s_h = (x, y_{1:h-1})$ includes the prompt $x$ and all response tokens produced up to that point. Each action $a_h = y_h$ represents a token from the vocabulary. The transition kernel $\mathcal{P}$ is usually known and deterministic, meaning that given tokens $s_h = (x, y_{1:h-1})$ and $a_h = y_h$, the environment will transition to $s_{h+1} = (x, y_{1:h})$. The policy $\pi$ maps all the observed tokens so far to a distri-

bution over the vocabulary. It is important to note that the policy captures the autoregressive nature of LLMs, i.e., $\pi(y_{1:h} \,|\, x) = \prod_{i=1}^{h} \pi(y_i \,|\, x, y_{1:h-1})$ for any $h$. Due to this, we may refer to it as an autoregressive policy to differentiate it from policies defined in other ways. Moreover, $r : \mathcal{S} \times \mathcal{A} \to \mathbb{R}$ represents the token-wise reward. The maximum number of tokens that can be generated, $H$, characterizes the length limit for LLM outputs. Each generated text ends with a special end-of-sentence token $\texttt{EoS}$, which terminates the generation process.

In our MDP formulation for RLHF, we also model the preference signal using BT model (Bradley & Terry, 1952), but replace the sentence-level reward function in (2.1) with token-wise reward functions. In specific, for any trajectory pair $\tau^1 = \{(s_h^1, a_h^1)\}_{h=1}^{H}$ and $\tau^2 = \{(s_h^2, a_h^2)\}_{h=1}^{H}$[1], the preference is specified by

$$\mathbb{P}(\tau^1 \succ \tau^2) = \sigma\bigg( \sum_{h=1}^{H} r(s_h^1, a_h^1) - \sum_{h=1}^{H} r(s_h^2, a_h^2) \bigg). \quad (3.1)$$

Compared to literature that formulates the RLHF problem as a contextual dueling bandit, a subtle difference is that the policy in the contextual dueling bandit maps a prompt to a distribution over sentences, which does not capture the autoregressive nature of LLMs. In contrast, our MDP formulation precisely captures this nature. We defer the discussion of these two types of policies in Section C.2. More importantly, the main difference is that the reward function in the MDP formulation is defined on a token level, which contrasts significantly with the sentence-level reward in the contextual dueling bandit. We discuss the advantages of token-level rewards in Section 3.3.

### 3.2. Learning Objective

Different from classical RL literature, where the sole goal is to maximize the reward function, the objective of RLHF is to maximize the reward function while ensuring that the learned policy does not deviate too much from the reference model (e.g., SFT model) too much. Inspired by this and the formulation of entropy-regularized MDPs (Williams & Peng, 1991; Ziebart, 2010), for any policy $\pi$, we define its corresponding regularized value-function by

$$V_\beta^\pi(s; r) = \mathbb{E}_\pi\bigg[ \sum_{h=1}^{\infty} \bigg( r(s_h, a_h) - \beta \cdot \log \frac{\pi(a_h \,|\, s_h)}{\pi_{\text{ref}}(a_h \,|\, s_h)} \bigg) \bigg| s_1 = s \bigg], \quad (3.2)$$

where the expectation $\mathbb{E}_\pi$ is taken with respect to the randomness incurred by the policy $\pi$. Here the summation ends

---

[1]In fact, these two trajectories can have different lengths, say $\tau^1 = \{(s_h^1, a_h^1)\}_{h=1}^{H_1}$ and $\tau^2 = \{(s_h^2, a_h^2)\}_{h=1}^{H_2}$ with $1 \le H_1, H_2 \le H$. These trajectories can be extended to length $H$ by assuming that the state ending with $\texttt{EoS}$ is absorbing and yields zero reward. This modification is to simplify the mathematical formulation and does not affect the problem modeling in (3.1). For the sake of clarity, the following theoretical discussion may focus on length-$H$ trajectories.

when a certain condition is met. In particular, since we assume that the maximal length of the generated responses of LLMs is at most $H$, the summation in (3.2) is taken at most $H$ steps. In the remaining part of this paper, we may use $\sum_{h=1}^{\infty}$ and $\sum_{h=1}^{H}$ interchangeably, as they mostly have the same meaning. The regularized Q-function $Q_\beta^\pi$ of a policy $\pi$ is related to the regularized value function $V_\beta^\pi$ as

$$Q_\beta^\pi(s, a; r) = r_\beta(s, a) + \mathbb{E}_{s' \sim \mathcal{P}(\cdot \,|\, s, a)}[V_\beta^\pi(s'; r)], \quad (3.3)$$
$$V_\beta^\pi(s; r) = \mathbb{E}_{a \sim \pi(\cdot \,|\, s)}[-\beta \log \pi(a \,|\, s) + Q_\beta^\pi(s, a; r)],$$

where we denote $r_\beta(s, a) = r(s, a) + \beta \log \pi_{\text{ref}}(a \,|\, s)$. Moreover, when it is clear from the context, we may omit the dependency of the ground-truth reward function $r$ in $Q_\beta^\pi(s, a; r), V_\beta^\pi(s; r)$ and use the shorthand $Q_\beta^\pi(s, a), V_\beta^\pi(s)$. The regularized optimal policy $\pi_\beta^*$ is the policy that maximizes the regularized value function defined in (3.2), and its corresponding optimal Q-function and value function are denoted as $Q_\beta^*$ and $V_\beta^*$, respectively. By (3.3), it can be shown that

$$\pi_\beta^*(a \,|\, s) = \exp\{(Q_\beta^*(s, a) - V_\beta^*(s))/\beta\}. \quad (3.4)$$

Our learning objective is to find a near-optimal policy $\widehat{\pi}$, whose performance is measured by the following suboptimality gap:

$$\text{SubOpt}(\widehat{\pi}) = V_\beta^*(\rho) - V_\beta^{\widehat{\pi}}(\rho), \quad (3.5)$$

where we use the shorthand $V_\beta^\pi(\rho) = \mathbb{E}_{s \sim \rho}[V_\beta^\pi(s)]$ for any policy $\pi$. For ease of presentation, we define the state visitation measure $d^\pi(s) = \mathbb{E}_{s_1 \sim \rho}[\sum_{h=1}^{\infty} \mathbb{P}(s_t = s \,|\, s_1)]$ and the state-action visitation measure $d^\pi(s, a) = \mathbb{E}_{s_1 \sim \rho}[\sum_{h=1}^{\infty} \mathbb{P}(s_h = s, a_h = a \,|\, s_1)]$. We also use the shorthand $d^* = d^{\pi_\beta^*}$ to further simplify the notation.

### 3.3. Advantages of Token-Wise MDP over Sentence-Wise Bandit

Intuitively, the distinction between token-based and trajectory-based rewards reflects the difference between sparse and dense reward settings. In the sparse reward scenario, exploration proves to be more challenging. To illustrate this, we focus on the deterministic MDP with an action set size of $A = |\mathcal{A}|$. We employ an autoregressive policy $\pi^*$ to represent the policy of a powerful LLM, such as GPT-4. Fixing a prompt $x$, given responses $(y^1 = y_{1:H}^1, y^2 = y_{1:H}^2)$, the evaluation provided by $\pi^*$ is $\mathbb{P}(y^1 \succ y^2 \,|\, x, y_1, y_2) = \frac{\pi^*(y^1 \,|\, x)}{\pi^*(y^1 \,|\, x) + \pi^*(y^2 \,|\, x)}$. By comparing this with the BT models of bandit in (2.1) and of our MDP formulation in (3.1), we observe that the sentence-wise reward $r_s$ and token-wise as $r_t$ can be specified by $\mathbf{r_s}(\mathbf{x}, \mathbf{y}) = \log \pi^*(y \,|\, x)$ and $\mathbf{r_t}((\mathbf{x}, \mathbf{y_{1:h-1}}), \mathbf{y_h}) = \log \pi^*(y_h \,|\, x, y_{1:h-1})$. Intuitively, the responses that powerful LLMs tend to choose have higher rewards. It is straightforward to show that $r_s(x, y) = \sum_{h=1}^{H} r_t((x, y_{1:h-1}), y_h)$. We make the following natural assumption.

**Algorithm 1** Reinforced Token Optimization (Theoretical Version)

1: **Input:** Offline dataset $\mathcal{D}$, $\lambda > 0$, $\beta > 0$, and problem dependent coefficient $\varrho$.
2: Compute $\theta_{\mathrm{MLE}}$ based on $\mathcal{D}$ by maximizing the loglikelihood given in (4.1).
3: Calculate the pessimistic reward $\widehat{r}$ via (4.2).
4: Compute the corresponding optimal policy $\widehat{\pi}$ with respect to $\widehat{r}$.
5: **Output:** policy $\widehat{\pi}$.

**Assumption 3.1.** There exists a response $y = y_{1:H}$ satisfying $\pi^*(y \mid x) \geq A^{-\xi}$ for some $0 \leq \xi \leq H$.

By the pigeon-hole principle, there must be a response $y$ such that $\pi^*(y \mid x) \geq A^{-H}$, implying that $\xi \leq H$. In practice, $\xi$ is usually much smaller than $H$ because the language model tends to choose the optimal response rather than making a random guess. Now, we define the interaction protocol and the sample complexity. The learner can determine a response $y = y_{1:H}$ and receive either $r_s(x, y)$ or $\{r_t((x, y_{1:h-1}), y_h)\}_{h=1}^H$, depending on whether the sentence-level reward or the token-wise reward is used. The sample complexity is defined as the number of responses and corresponding reward signals that need to be gathered to find the optimal response $y^* = y_{1:H}^*$ with length $H$.

**Proposition 3.2.** *Suppose Assumption 3.1 holds. In the setting where only the sentence-wise reward $r_s$ is accessible, finding the optimal response $y^*$ requires a sample complexity of $A^H$. However, if token-reward signals $r_t$ are available, there exists an algorithm that can find the optimal policy with sample complexity $A^{\min\{\xi+1, H\}}$.*

The proof is deferred to Appendix A.1. Since $\xi \ll H$ typically holds in practice, the gap between $A^H$ and $A^{\min\{\xi+1, H\}}$ is deemed large. Hence, Proposition 3.2 reveals the significant separation of sample complexity between two types of reward signals, providing theoretical insights into the superiority of the token-wise MDP formulation over the sentence-wise bandit formulation.

# 4. Reinforced Token Optimization

We develop the Reinforced Token Optimization (RTO) algorithm under the MDP framework introduced in Section 3. At a high level, RTO consists of two main steps: (i) token-wise reward learning, where RTO learns a token-wise reward based on the preference data; and (ii) optimizing token-wise reward through RL training methods such as PPO. In Section 4.1, we provide a theoretically grounded version of RTO with guaranteed sample complexity. Then we present a practical implementation of RTO in Section 4.2.

## 4.1. Theoretical Version with Sample Complexity Guarantee

We focus on the offline setting and assume the access to an offline dataset $\mathcal{D} = \{(\tau^w, \tau^l)\}$ that contains several trajectory pairs, where $\tau^w = \{(s_h^w, a_h^w)\}_{h=1}^H$ is preferred over $\tau^l = \{(s_h^l, a_h^l)\}_{h=1}^H$. Each pair of trajectories shares the same initial state/prompt (i.e., $s_1^w = s_1^l$), but differs in the subsequent tokens. We also assume that the reward function is linear, and our following results are ready to be extended to general function approximation (Chen et al., 2022; Wang et al., 2023; Zhan et al., 2023a).

**Assumption 4.1** (Linear Reward). We assume $r(s, a) = \phi(s, a)^\top \theta^*$ for some known feature $\phi : \mathcal{S} \times \mathcal{A} \to \mathbb{R}^d$ and unknown vector $\theta^* \in \mathbb{R}^d$. We also assume that $\|\phi(\cdot, \cdot)\|_2 \leq L$ and $\|\theta^*\|_2 \leq B$.

Following the standard reward learning pipeline (Ouyang et al., 2022), we learn the reward function via maximum likelihood estimation (MLE). Specifically, if we parametrize the reward function by $\theta$, then the MLE is given by

$$\theta_{\mathrm{MLE}} = \underset{\|\theta\|_2 \leq B}{\mathrm{argmax}} \, \mathcal{L}_{\mathcal{D}}(\theta), \qquad (4.1)$$

where $\mathcal{L}_{\mathcal{D}}(\theta) = \sum_{(\tau^w, \tau^l) \in \mathcal{D}} [\log(\sigma(\sum_{h=1}^H r_\theta(s_h^w, a_h^w) - \sum_{h=1}^H r_\theta(s_h^l, a_h^l)))]$. Inspired by previous literature in offline RL (Jin et al., 2021; Rashidinejad et al., 2021; Xiong et al., 2022; Zhu et al., 2023; Zhan et al., 2023a), given the MLE $\theta_{\mathrm{MLE}}$, we construct the pessimistic token-wise reward estimation as

$$\widehat{r}(s, a) = \phi(s, a)^\top \theta_{\mathrm{MLE}} - \varrho \cdot \|\phi(s, a)\|_{\Sigma_{\mathcal{D}}^{-1}}, \qquad (4.2)$$

where $\Sigma_{\mathcal{D}} = \sum_{(\tau^1, \tau^2) \in \mathcal{D}} [\sum_{h=1}^H (\phi(s_h^1, a_h^1) - \phi(s_h^2, a_h^2))(\sum_{h=1}^H (\phi(s_h^1, a_h^1) - \phi(s_h^2, a_h^2)))^\top] + \lambda I_d$, $\lambda > 0$ is a tuning parameter, and $\varrho$ is a problem-dependent coefficient will be specified in Theorem 4.2 and (A.3). Finally, RTO outputs the optimal policy $\widehat{\pi}$ with respect to $\widehat{r}$, i.e., $\widehat{\pi} = \mathrm{argmax}_\pi V_\beta^\pi(s; \widehat{r})$ for any $s \in \mathcal{S}$. The pseudocode of RTO is given in Algorithm 1.

**Theorem 4.2.** *Suppose Assumption 4.1 holds. For $\beta > 0$, $\lambda > 0$, $\delta \in (0, 1)$, if we choose $\varrho = \widetilde{\mathcal{O}}(\sqrt{d})$ (see (A.3)), then the output policy $\widehat{\pi}$ of Algorithm 1 satisfies*

$$\mathrm{SubOpt}(\widehat{\pi}) \leq 2\varrho \cdot \mathbb{E}_{(s,a) \sim d^*} \big[ \|\phi(s, a)\|_{\Sigma_{\mathcal{D}}^{-1}} \big]$$
$$- \beta \cdot \mathbb{E}_{s \sim d^*} \big[ \mathrm{KL}\big(\pi_\beta^*(\cdot \mid s) \| \widehat{\pi}(\cdot \mid s)\big) \big].$$

The detailed proof is deferred to Appendix A.2. The first term in Theorem 4.2 measures how well the offline dataset covers the trajectory generated by the policy $\pi_\beta^*$. Typically, this term decreases at a rate of $|\mathcal{D}|^{-1/2}$ under the mild partial coverage assumption (Jin et al., 2021; Uehara & Sun, 2021; Xiong et al., 2022; Zhu et al., 2023; Zhan et al., 2023a),

**Algorithm 2** Reinforced Token Optimization (Practical Version)

---

1: **Input:** Offline dataset $\mathcal{D}$, parameters $\beta_1, \beta_2 > 0$, DPO algorithm `DPO`, and PPO trainer `PPO-Update`.
2: Compute $\pi_{\text{dpo}} \leftarrow \text{DPO}(\mathcal{D})$ and let $\pi_0 = \pi_{\text{ref}}$ as the reference model.
3: **for** $t = 1, \ldots, T$ **do**
4:     Get a batch of samples $\mathcal{D}_t$ from the dataset $\mathcal{D}$ but we only keep the prompts.
5:     For each prompt $x \in \mathcal{D}_t$, generate a response $y \sim \pi_{t-1}(\cdot \mid x)$.
6:     Calculate the token-wise reward $r_{\text{rto}}$ by (4.7).
7:     $\pi_t \leftarrow \text{PPO-Update}(\pi_{t-1}, r_{\text{rto}}, \{(x,y)\}_{x \in \mathcal{D}_t})$.
8: **end for**
9: **Output:** policy $\pi_T$.

---

where $|\mathcal{D}|$ is the size of the offline dataset. The second KL term is always negative, and it arises from the goal of learning a regularized value. We also remark that our algorithm relies on the known transition kernel to compute the exact optimal policy with respect to $\widehat{r}$. While this is natural in the context of large language models, we provide insights on how to extend our findings to stochastic regularized MDPs and the variant of our `RTO` algorithm in Appendix B.

There have also been previous works (Pacchiano et al., 2021; Chen et al., 2022; Wang et al., 2023; Li et al., 2023c; Zhan et al., 2023a) studying RLHF under the MDP framework, also known as dueling RL and preference-based RL. However, these works do not consider the KL constraint, which is an essential component of RLHF. Furthermore, they do not explicitly emphasize the superiority of the MDP framework over the contextual dueling bandit problem in the context of LLMs, and their proposed algorithms does not lead to corresponding practical algorithm leveraging dense rewards. In contrast, we will provide a practical implementation of our algorithm, demonstrating the practicality of our approach.

### 4.2. Practical Implementation

In this subsection, we shift our focus to developing a practical version of `RTO`. The key challenge in implementing `RTO` in Algorithm 1 lies in learning the token-wise reward to be optimized from the offline data. In the most popular frameworks outlined in Instruct-GPT (Ouyang et al., 2022), Claude (Bai et al., 2022), and LLaMA2 (Touvron et al., 2023) projects replace the last layer of the LLM with a linear layer for a scalar output and maximize the log-likelihood as in (2.2). However, this approach gives only a sentence-level reward. To bridge the gap in the literature, we present our practical version of `RTO` in Algorithm 2, which features a novel calculation of token-wise reward. Our key observation is that, given a trajectory $\tau = \{(s_h, a_h)\}_{h=1}^H$, we can

rewrite $\sum_{h=1}^H \beta \log \frac{\pi_\beta^*(a_h \mid s_h)}{\pi_{\text{ref}}(a_h \mid s_h)}$ as

$$\sum_{h=1}^H \left( Q_\beta^*(s_h, a_h) - V_\beta^*(s_h) - \log \pi_{\text{ref}}(a_h \mid s_h) \right)$$

$$= \sum_{h=1}^H r(s_h, a_h) - V_\beta^*(s_1)$$

$$+ \underbrace{\sum_{h=1}^{H-1} \left( \mathbb{E}_{s' \sim \mathcal{P}(\cdot \mid s_h, a_h)}[V_\beta^*(s')] - V_\beta^*(s_{h+1}) \right)}_{(\star)}, \quad (4.3)$$

where the first equality uses the closed-form of optimal policy $\pi_\beta^*(a \mid s) = \exp\{(Q_\beta^*(s,a) - V_\beta^*(s))/\beta\}$ in (3.4), and the second equality follows from the fact that $Q_\beta^\pi(s,a) = r_\beta(s,a) + \mathbb{E}_{s' \sim \mathcal{P}(\cdot \mid s,a)}[V_\beta^\pi(s')]$ in (3.3) with $r_\beta(s,a) = r(s,a) + \beta \log \pi_{\text{ref}}(a \mid s)$. We focus on the typical LLM generation scenario where the transition kernel is deterministic. Then we have $(\star) = 0$ in (4.3), yielding that $\sum_{h=1}^H r(s_h, a_h) = \sum_{h=1}^H \beta \log \frac{\pi_\beta^*(a_h \mid s_h)}{\pi_{\text{ref}}(a_h \mid s_h)} + V_\beta^*(s_1)$. Building upon this result and combining it with the definition of the BT model in (3.1), for any trajectory pair $\{\tau^j = \{(s_h^j, a_h^j)\}_{h=1}^H\}_{j=1}^2$ satisfying $s_1^1 = s_1^2$, we have

$$\mathbb{P}(\tau^1 \succ \tau^2) = \sigma\left( \sum_{h=1}^H r(s_h^1, a_h^1) - \sum_{h=1}^H r(s_h^2, a_h^2) \right) \quad (4.4)$$

$$= \sigma\left( \sum_{h=1}^H \beta \log \frac{\pi_\beta^*(a_h^1 \mid s_h^1)}{\pi_{\text{ref}}(a_h^1 \mid s_h^1)} - \sum_{h=1}^H \beta \log \frac{\pi_\beta^*(a_h^2 \mid s_h^2)}{\pi_{\text{ref}}(a_h^2 \mid s_h^2)} \right).$$

An interesting observation is that, based on the autoregressive nature of policies, (4.4) aligns with the learning objective of DPO proposed by Rafailov et al. (2023), but under the token-level MDP instead of the sentence-level bandit setup. Similar to the bandit setting where the learning objective is equivalent to a BT model with sentence-wise reward $r^*(x, y) = \beta \log \frac{\pi_\beta^*(y \mid x)}{\pi_{\text{ref}}(y \mid x)}$ (Rafailov et al., 2023), (4.4) shows that the learning objective in token-wise MDP equivalents to a BT model with a token-wise reward function

$$r^*(s_h = (x, y_{1:h-1}), a_h = y_h)$$

$$= \beta \log \frac{\pi_\beta^*(a_h \mid s_h)}{\pi_{\text{ref}}(a_h \mid s_h)} = \beta \log \frac{\pi_\beta^*(y_h \mid x, y_{1:h-1})}{\pi_{\text{ref}}(y_h \mid x, y_{1:h-1})}, \quad (4.5)$$

where $x$ is the prompt, $y_{1:h-1}$ is the tokens generated so far, and $y_h$ is the token chosen at the current step. In contrast to the previous PPO implementation with sparse reward in (2.3), we will assign the token-wise reward function defined in (4.5) to each step. Formally, for any $h$, we define

$$\beta_1 \log \frac{\pi_\beta^*(y_h \mid x, y_{1:h-1})}{\pi_{\text{ref}}(y_h \mid x, y_{1:h-1})} - \beta_2 \log \frac{\pi(y_h \mid x, y_{1:h-1})}{\pi_{\text{ref}}(y_h \mid x, y_{1:h-1})}$$

$$\approx \beta_1 \log \frac{\pi_{\text{dpo}}(y_h \mid x, y_{1:h-1})}{\pi_{\text{ref}}(y_h \mid x, y_{1:h-1})} - \beta_2 \log \frac{\pi(y_h \mid x, y_{1:h-1})}{\pi_{\text{ref}}(y_h \mid x, y_{1:h-1})}$$

$$\quad (4.6)$$

| Metric | Method | | | | | | |
|--------|--------|--------|--------|--------|--------|--------|--------|
|        | SFT | DPO | R-DPO | SimPO | TDPO | PPO | RTO |
| AE (LC) | 13.22 | 17.40 | 18.34 | 25.46 | 20.13 | 19.47 | **27.00** |
| AE (WR) | 8.58 | 12.23 | 12.03 | 20.20 | 11.97 | 12.89 | **22.45** |
| AH (SC) | 9.2 | 13.2 | 14.2 | 14.5 | 13.2 | 16.2 | **20.3** |
| AH (WR) | 8.9 | 13.8 | 14.1 | 15.2 | 12.3 | 15.6 | **21.4** |

*Table 1.* AlpacaEval 2 (**AE**) and Arena-Hard (**AH**) results.

| Metric | Method | | | |
|--------|--------|----------|--------|--------|
|        | RTO | Semi-RTO | DPPO | RS-PPO |
| AE (LC) | 27.00 | 23.77 | 21.09 | **27.52** |
| AE (WR) | **22.45** | 19.17 | 13.06 | 21.69 |
| AH (SC) | **20.3** | 19.0 | 13.1 | 19.2 |
| AH (WR) | **21.4** | 19.7 | 12.1 | 19.9 |

*Table 2.* Benchmark performance of ablations studies.

as the token-wise reward, where $\beta_1$ and $\beta_2$ are tuning parameters, and $\pi$ is the current policy to be updated. In the last step of (4.6), we use $\pi_{\mathrm{dpo}}$, the policy learned by DPO, as a proxy for the unknown $\pi_\beta^*$. Finally, we employ PPO to optimize the following token-wise reward $r_{\mathrm{rto}}$

$$r_{\mathrm{rto}}(x, y_{1:h}) \qquad\qquad (4.7)$$
$$= \begin{cases} \beta_1 \log \frac{\pi_{\mathrm{dpo}}(y_h \mid x, y_{1:h-1})}{\pi_{\mathrm{ref}}(y_h \mid x, y_{1:h-1})} - \beta_2 \log \frac{\pi(y_h \mid x, y_{1:h-1})}{\pi_{\mathrm{ref}}(y_h \mid x, y_{1:h-1})} \\ \qquad\qquad \text{if } h \le H - 1, \\ \beta_1 \log \frac{\pi_{\mathrm{dpo}}(y_h \mid x, y_{1:h-1})}{\pi_{\mathrm{ref}}(y_h \mid x, y_{1:h-1})} - \beta_2 \log \frac{\pi(y_h \mid x, y_{1:h-1})}{\pi_{\mathrm{ref}}(y_h \mid x, y_{1:h-1})} \\ \qquad + \beta_3 \cdot r_{\mathrm{MLE}}(x, y_{1:H}) \quad \text{if } h = H, \end{cases}$$

where $\beta_3 \ge 0$ is a tuning parameter and $r_{\mathrm{MLE}}$ represents a sentence-level reward. This additional sentence-level reward helps prevent responses from becoming either extremely long or extremely short. This aligns with the observation that ensemble rewards can effectively mitigate the overoptimization issues (Coste et al., 2023). We also remark that the sentence-level reward $r_{\mathrm{MLE}}$ can be much smaller in magnitude compared to both the policy model (actor) and DPO reward model, making the overall computational cost of RTO *comparable* to the standard RLHF pipeline: The lower cost of using a much smaller critic in PPO compensates for both the small extra cost required by DPO than reward model, and the training and serving of the tiny reward model. For parameter selection, $\beta_3$ can be set to 1 when $r_{\mathrm{MLE}}$ is included. This choice is without loss of generality, as the key factor is the ratio of $\beta_3$ to $\beta_1$ and $\beta_2$, rather than its absolute value. $\beta_2$ can be chosen similarly in standard PPO configurations. The only extra hyperparameter $\beta_1$ can be set small to prevent the DPO reward from dominating, thereby requiring minimum tuning.

## 5. Experiments

### 5.1. Benchmark Results

We present a thorough comparison of RTO with PPO and other widely used direct preference learning algorithms on popular benchmarks to highlight RTO's strong performance.

**Task, Data, and Evaluation.** To assess the overall quality of generated text responses across multiple dimensions (e.g., helpfulness, accuracy, and clarity), we employ the dataset UltraFeedback (Cui et al., 2023) that contains comprehensive human feedback annotations on model outputs. We evaluate models using two established benchmarks: Al-

pacaEval 2 (Li et al., 2023a) and Arena-Hard (Li et al., 2024). These benchmarks assess various conversational abilities across different types of queries. For AlpacaEval 2, we report both standard win rates (WR) and length-controlled win rates (LC). For Arena-Hard, we present the WR along with its style-controlled (SC) version. Both LC and SC are specifically designed to mitigate verbosity bias of llm judge.

**Implementation Details of RTO and Baselines.** We employ Llama-3-8B (Dubey et al., 2024) as the base model. For our comparative analysis, we implement several baselines. All subsequent models are initialized with an open-source **SFT** model (Dong et al., 2024) that fine-tunes LLama-3-8B with a diverse mixture of high-quality data. We further train a **DPO** model, which finetunes the SFT model using the positive/negative preference data. Besides these two RL-free algorithms, we compare three RLHF algorithms relying on RL training. The first one is the standard **PPO** algorithm, which directly optimizes sentence-level 8B reward in (2.3). Our proposed **RTO** algorithm leverages both token-wise signals from the DPO model and an additional 1B sentence-wise reward $r_{\mathrm{MLE}}$ to compute the RTO reward specified in (4.7). The policy is then trained to align with human preferences using PPO updates, as detailed in Algorithm 2. For a comprehensive comparison, we include the following methods as baselines: (i) **R-DPO** (Park et al., 2024) is a variant of DPO that regularizes response length, (ii) **SimPO** (Meng et al., 2024) optimizes average likelihood directly and (iii) **TDPO** (Zeng et al., 2024) reformulates and optimizes DPO at token-level. We include more implementation details in appendix D.

**RTO Outperforms PPO and Other Direct Preference Learning Algorithms.** As demonstrated in Table 1, while all evaluated algorithms show improvement over the base SFT model, our RTO algorithm achieves superior performance across all benchmarks. Specifically, RTO outperforms PPO by achieving a 7.53% higher win rate in the AlpacaEval 2 LC benchmark and a 4.1% higher win rate in the Arena-Hard SC benchmark. These results highlight the effectiveness of incorporating token-wise reward (dense reward) into PPO training. Furthermore, when compared to the leading preference learning baselines, RTO demonstrates improvements of 2 and 4 points in the AlpacaEval 2 LC benchmark and the Arena-Hard SC benchmark, respectively.

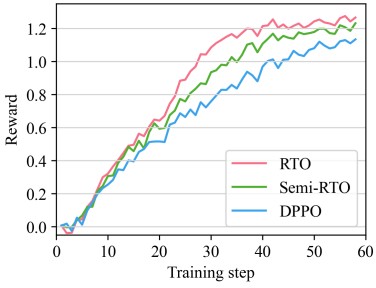
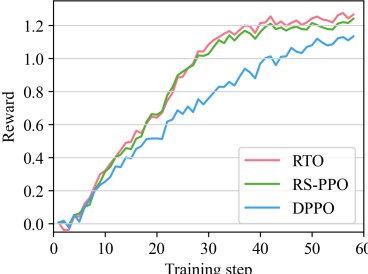
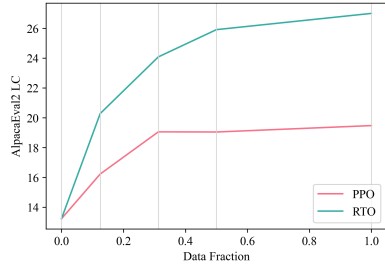

(a) Effect of different reward granularity.     (b) Effect of reward shaping.     (c) Data scaling behavior of PPO and RTO.

*Figure 2.* (a) and (b) show the obtained reward of $r_{\mathrm{MLE}}$ throughout training. (c) shows the AlpacaEval 2 performance of PPO and RTO when trained on fractions of samples.

### 5.2. In-depth Analysis of RTO Performance

In this subsection, we provide a detailed analysis of RTO. First, we examine the influence of reward granularity by comparing the performance of RL training based on different reward types. Next, we demonstrate that the token-wise reward (provided by DPO) primarily serves as reward shaping, which contributes significantly to RTO's success. Finally, we investigate RTO's sample efficiency to support our theoretical findings.

**The Influence of Reward Granularity.** We analyze three reward granularity settings by redistributing token-level rewards used in RTO: (i) **RTO**, where rewards are assigned to each token; (ii) **Semi-RTO**, where the rewards of all tokens in each sentence are reassigned to their delimiter, and (iii) **DPPO**, where all rewards are delayed and assigned to the EoS token. Table 2 and Figure 2(a) clearly demonstrates that denser rewards lead to better performance.

**Reward Shaping via DPO Reward is the Key to RTO's Success.** We demonstrate that the superior performance of RTO is not primarily due to replacing the reward model trained with MLE with the implicit reward from DPO. Instead, its advantage lies in its role as a reward-shaping mechanism. To illustrate this, we compare RTO and DPPO with another setup, **RS-PPO**, where the reward matches $r_{\mathrm{rto}}$ in (4.7), except for subtracting the DPO implicit reward $\beta_1 \log \frac{\pi_{\mathrm{dpo}}(y|x)}{\pi_{\mathrm{ref}}(y|x)}$ from the last token. This adjustment results in a total reward equivalent to $r_{\mathrm{MLE}}(x, y)$, effectively employing the DPO token-wise implicit reward for reward shaping. From Table 2 and Figure 2(b), we observe that the main contribution of the DPO reward to improving RL training lies in reward shaping rather than altering the total reward through its exact value.

**Sample Efficiency of RTO.** To validate the theoretical claims in Theorem 4.2, which guarantees the provable sample efficiency of RTO, and Proposition 3.2, which demonstrates that RTO is more efficient than PPO due to its use of token-wise rewards instead of sentence-wise rewards, we conducted experiments using only a fraction of the full

dataset. These experiments evaluated the ability of RTO and PPO to learn an effective policy with limited data. As shown in Figure 2(c), RTO matches PPO's performance using only about $1/8$ of the data and ultimately surpasses PPO's final performance. Additionally, RTO exhibits superior data scaling behavior compared to PPO — RTO continues to improve with more data, while PPO's performance saturates early.

**Additional Experiments.** To showcase the applicability of RTO, we conducted additional experiments demonstrating: (i) its effectiveness **beyond PPO** by incorporating the learned token-wise reward function into REINFORCE-type algorithms (Williams, 1992; Hu, 2025), yielding significant improvements (Appendix E), and (ii) its utility for diverse alignment tasks beyond dialogue, such as text **summarization** (Völske et al., 2017) (Appendix F).

## 6. Conclusion

In this work, we propose an MDP formulation for RLHF that better characterizes token-wise information, along with theoretical insights demonstrating its superiority. Building upon this formulation, we introduce a novel algorithm called Reinforced Token Optimization (RTO), which leverages token-wise rewards to improve the policy. RTO is shown to be both provably sample-efficient and practical. Our practical implementation involves a novel token-wise reward learning approach via DPO, followed by optimization using PPO. This innovative combination of DPO and PPO allows RTO to effectively utilize token-level information and significantly improve the performance of baselines. Furthermore, our research opens up several intriguing future directions, such as designing alternative methods for learning token-wise rewards beyond DPO.

## Impact Statement

This paper presents work whose goal is to advance the field of Machine Learning. There are many potential societal consequences of our work, none which we feel must be specifically highlighted here.

## Acknowledgement

LW is supported by National Science and Technology Major Project (2022ZD0114902) and National Science Foundation of China (NSFC92470123, NSFC62276005). DH is supported by National Science Foundation of China (NSFC62376007). This work is partially supported by the Shanghai Committee of Science and Technology (Grant No.21DZ1100100).

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

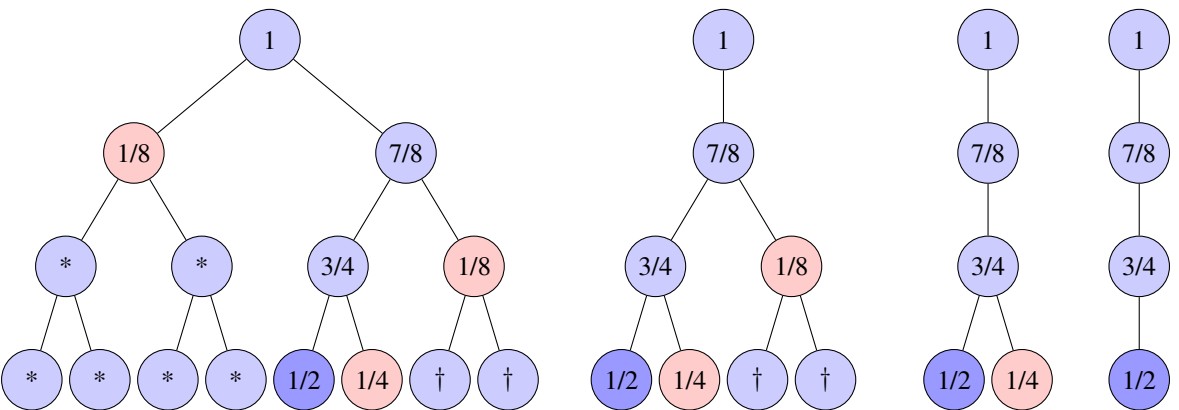

*Figure 3.* An illustration of our efficient learning algorithm for the token-wise reward setting with $A = 2$, $H = 3$, and $\xi = 1$. Here $*$ and $\dagger$ represent real numbers between 0 and 1/8. We do not specify their exact values as they do not influence the optimal path. All nodes in $\mathcal{N}$ are colored red, while other nodes are blue, with the optimal leaf node 1/2 emphasized in dark blue. Each node $y_{1:h}$ is labelled with $\pi^*(y_{1:h} \mid x)$. If a non-optimal path (response) is selected, one red node in $\mathcal{N}$ will be identified, and all paths containing this node will be deleted. Here we visualize the process of choosing a path ending with $*$, $\dagger$, and 1/4, respectively. At most $A^{\min\{\xi+1, H\}} = 4$ samples are needed to identify the optimal response.

## A. Detailed Proofs

### A.1. Proof of Proposition 3.2

*Proof.* If only the sentence-level reward $r_s$ is available, the learner must try every possible response and determine the optimal one by ranking the collected sentence-level reward signals, resulting in a sample complexity of $A^H$. Instead, we consider a binary tree with depth $H + 1$, where each node is indexed by some token sequence $y_{1:h}$ and has $A$ children $\{(y_{1:h}, y_{h+1})\}_{y_{h+1} \in \mathcal{A}}$. All $A^H$ leaf nodes denote a unique prompt-response pair $(x, y_{1:H})$. We define two disjoint node sets:

$$\mathcal{N} = \left\{ y_{1:h} : \pi^*(y_{1:h} \mid x) < A^{-\xi}, \pi^*(y_{1:h-1} \mid x) \geq A^{-\xi} \right\}, \qquad \mathcal{N}^* \left\{ y_{1:H} : \pi^*(y_{1:H} \mid x) \geq A^{-\xi} \right\}. \tag{A.1}$$

Our key observations are that (i) each path must contain a node in $\mathcal{N}$ or $\mathcal{N}^*$, (ii) the path containing the node in $\mathcal{N}$ is suboptimal; and (iii) $|\mathcal{N} \cup \mathcal{N}^*| \leq A^{\xi+1}$. The exploration strategy is to query a new path that does not contain the nodes in $\mathcal{N} \cup \mathcal{N}'$ that have been visited. Since each query of a new path (response with length $H$) can identify a new additional node in $\mathcal{N} \cup \mathcal{N}^*$, after at least $A^{\xi+1}$ queries, we collect a set of paths where each node in $\mathcal{N} \cup \mathcal{N}^*$ belongs to one of the paths. Finally, ranking all gathered rewards of the node in $\mathcal{N}^*$ identifies the optimal $y^* = y_{1:H}^*$. Together with the fact that there exists as most $A^H$ nodes, we finish the proof of Theorem 3.2. To facilitate understanding, we visualize a simplified learning process in Figure 3. $\qquad\square$

### A.2. Proof of Theorem 4.2

Recall that the visitation measure of policy $\pi$ is

$$d^\pi(s) = \mathbb{E}_{s_1 \sim \rho}\left[ \sum_{h=1}^\infty \mathbb{P}(s_t = s \mid s_1) \right], \qquad d^\pi(s, a) = \mathbb{E}_{s_1 \sim \rho}\left[ \sum_{h=1}^\infty \mathbb{P}(s_h = s, a_h = a \mid s_1) \right]. \tag{A.2}$$

Under this notation, we can rewrite the value function in (3.2) as

$$V_\beta^\pi(\rho) = \mathbb{E}_{(s,a) \sim d^\pi}\left[ r(s, a) - \mathrm{KL}\big(\pi(\cdot \mid s) \| \pi_{\mathrm{ref}}(\cdot \mid s)\big) \right].$$

For simplicity, we will use the shorthand $d^* = d^{\pi_\beta^*}$.

*Proof of Theorem 4.2.* Our proof relies on the following standard MLE analysis.

**Lemma A.1** (MLE Analysis). *It holds with probability $1 - \delta$ that*

$$\|\theta_{\mathrm{MLE}} - \theta^*\|_{\Sigma_{\mathcal{D}}} \leq \varrho := C \cdot \sqrt{\frac{d \log(1/\delta)}{\Upsilon} + \lambda B^2}, \tag{A.3}$$

*where $C$ is an absolute constant and $\Upsilon = 1/(2 + \exp(-2HLB) + \exp(2HLB))$.*

*Proof.* See e.g., Faury et al. (2020); Pacchiano et al. (2021); Zhu et al. (2023) for a detailed proof. □

Back to the proof of Theorem 4.2, we first decompose the suboptimality gap defined in (3.5) as

$$\begin{aligned}
\mathrm{SubOpt}(\widehat{\pi}) &= V_\beta^*(\rho; r) - V_\beta^{\widehat{\pi}}(\rho; r) \\
&= \mathbb{E}_{(s,a) \sim d^*}\big[r(s,a) - \beta \cdot \mathrm{KL}\big(\pi_\beta^*(\cdot \,|\, s) \| \pi_{\mathrm{ref}}(\cdot \,|\, s)\big)\big] - \big(\mathbb{E}_{(s,a) \sim d^{\widehat{\pi}}}\big[r(s,a) - \beta \cdot \mathrm{KL}\big(\widehat{\pi}(\cdot \,|\, s) \| \pi_{\mathrm{ref}}(\cdot \,|\, s)\big)\big]\big) \\
&= \underbrace{\mathbb{E}_{(s,a) \sim d^*}[r(s,a) - \widehat{r}(s,a)]}_{\mathrm{Term(i)}} + \underbrace{\mathbb{E}_{(s,a) \sim d^{\widehat{\pi}}}[\widehat{r}(s,a) - r(s,a)]}_{\mathrm{Term(ii)}} + \underbrace{V_\beta^{\pi_\beta^*}(\rho; \widehat{r}) - V_\beta^{\widehat{\pi}}(\rho; \widehat{r})}_{\mathrm{Term(iii)}}.
\end{aligned} \tag{A.4}$$

Then we analyze these three terms respectively.

**Term (i).** Recall that the pessimistic reward $\widehat{r}$ defined in (4.2) takes the form

$$\widehat{r}(s,a) = \phi(s,a)^\top \theta_{\mathrm{MLE}} - \varrho \cdot \|\phi(s,a)\|_{\Sigma_{\mathcal{D}}^{-1}}.$$

Then we can rewrite Term (i) in (A.4) as

$$\begin{aligned}
\mathrm{Term(i)} &= \mathbb{E}_{(s,a) \sim d^*}\big[\phi(s,a)^\top(\theta^* - \theta_{\mathrm{MLE}}) + \varrho \cdot \|\phi(s,a)\|_{\Sigma_{\mathcal{D}}^{-1}}\big] \\
&\leq \mathbb{E}_{(s,a) \sim d^*}\big[\|\phi(s,a)\|_{\Sigma_{\mathcal{D}}^{-1}} \cdot \|\theta^* - \theta_{\mathrm{MLE}}\|_{\Sigma_{\mathcal{D}}} + \varrho \cdot \|\phi(s,a)\|_{\Sigma_{\mathcal{D}}^{-1}}\big] \\
&\leq 2\varrho \cdot \mathbb{E}_{(s,a) \sim d^*}\big[\|\phi(s,a)\|_{\Sigma_{\mathcal{D}}^{-1}}\big], \tag{A.5}
\end{aligned}$$

where the first inequality is obtained by Cauchy-Schwarz inequality, and the last inequality follows from Lemma A.1.

**Term (ii).** Similar to the derivation of (A.5), we have

$$\begin{aligned}
\mathrm{Term(ii)} &= \mathbb{E}_{(s,a) \sim d^{\widehat{\pi}}}\big[\phi(s,a)^\top(\theta_{\mathrm{MLE}} - \theta^*) - \varrho \cdot \|\phi(s,a)\|_{\Sigma_{\mathcal{D}}^{-1}}\big] \\
&\leq \mathbb{E}_{(s,a) \sim d^{\widehat{\pi}}}\big[\|\phi(s,a)\|_{\Sigma_{\mathcal{D}}^{-1}} \cdot \|\theta_{\mathrm{MLE}} - \theta^*\|_{\Sigma_{\mathcal{D}}} - \varrho \cdot \|\phi(s,a)\|_{\Sigma_{\mathcal{D}}^{-1}}\big] \\
&\leq 0, \tag{A.6}
\end{aligned}$$

where the first inequality uses Cauchy-Schwarz inequality, and the last inequality is implied by Lemma A.1.

**Term (iii).** To handle this term, we introduce the following performance difference lemma for MDP with KL constraint.

**Lemma A.2** (Performance Different Lemma). *For any reward function $r$ and policy pair $(\pi, \pi')$, it holds that*

$$V_\beta^\pi(\rho; r) - V_\beta^{\pi'}(\rho; r) = \mathbb{E}_{(s,a) \sim d^\pi}[Q_\beta^{\pi'}(s,a;r) - V_\beta^{\pi'}(s;r) - \beta \log \pi(a \,|\, s)].$$

*Proof.* See Appendix A.3 for a detailed proof. □

When $\beta = 0$, the regularized MDP becomes the standard MDP, and Lemma A.2 reduces to the standard performance difference lemma (Kakade & Langford, 2002). Applying Lemma A.2 to Term (iii) in (A.4), we have

$$\begin{aligned}
\mathrm{Term(iii)} &= \mathbb{E}_{(s,a) \sim d^*}[Q_\beta^{\widehat{\pi}}(s,a;\widehat{r}) - V_\beta^{\widehat{\pi}}(s;\widehat{r}) - \beta \log \pi_\beta^*(a \,|\, s)] \\
&= \mathbb{E}_{(s,a) \sim d^*}[\beta \log \widehat{\pi}(a \,|\, s) - \beta \log \pi_\beta^*(a \,|\, s)] \\
&= -\beta \cdot \mathbb{E}_{s \sim d^*}\big[\mathrm{KL}\big(\pi_\beta^*(\cdot \,|\, s) \| \widehat{\pi}(\cdot \,|\, s)\big)\big], \tag{A.7}
\end{aligned}$$

where the second equality follows from the fact that $\widehat{\pi}$ is the optimal policy with respect to $V_\beta^\pi(s; \widehat{r})$ and the expression of optimal policy $\widehat{\pi}(a \,|\, s) = \exp\{(Q_\beta^{\widehat{\pi}}(s,a;\widehat{r}) - V_\beta^{\widehat{\pi}}(s;\widehat{r}))/\beta\}$ in (3.4), and the last equality is obtained by the definition of KL divergence.

**Finishing the Proof.** Plugging (A.5), (A.6), and (A.7) into (A.4), we obtain that

$$\mathrm{SubOpt}(\widehat{\pi}) \le 2\varrho \cdot \mathbb{E}_{(s,a)\sim d^*}\big[\|\phi(s,a)\|_{\Sigma_{\mathcal{D}}^{-1}}\big] - \beta \cdot \mathbb{E}_{s\sim d^*}\big[\mathrm{KL}\big(\pi_\beta^*(\cdot\,|\,s)\|\widehat{\pi}(\cdot\,|\,s)\big)\big],$$

which finishes the proof of Theorem 4.2. $\qquad\square$

*Remark* A.3. If we do not have access to the exact optimal policy $\widehat{\pi}$ with respect to $\widehat{r}$, we can use the policy optimization algorithms to find a near-optimal optimal policy $\widetilde{\pi}$. In such case, Term (iii) in (A.6) becomes $V_\beta^{\pi_\beta^*}(\rho;\widehat{r}) - V_\beta^{\widetilde{\pi}}(\rho;\widehat{r}) = V_\beta^{\pi_\beta^*}(\rho;\widehat{r}) - V_\beta^{\widehat{\pi}}(\rho;\widehat{r}) + V_\beta^{\widehat{\pi}}(\rho;\widehat{r}) - V_\beta^{\widetilde{\pi}}(\rho;\widehat{r})$, and we need to handle the additional error term $V_\beta^{\widehat{\pi}}(\rho;\widehat{r}) - V_\beta^{\widetilde{\pi}}(\rho;\widehat{r})$. This type of error analysis has been established for NPG (Agarwal et al., 2021; Cen et al., 2022) and PPO (Cai et al., 2020; Wu et al., 2022; Zhong & Zhang, 2024).

## A.3. Proof of Lemma A.2

*Proof of Lemma A.2.* Without loss of generality, we assume that the initial state is a fixed state $s_1 \in \mathcal{S}$. For simplicity, we also omit the dependency of $r$ in the regularized Q-function and value function. First, we have

$$V_\beta^\pi(s_1) - V_\beta^{\pi'}(s_1) = \underbrace{V_\beta^\pi(s_1) - \mathbb{E}_{a_1\sim\pi(\cdot\,|\,s_1)}\big[r_\beta(s_1,a_1) + \mathbb{E}_{s_2\sim\mathcal{P}(\cdot\,|\,s_1,a_1)}[V_\beta^{\pi'}(s_2)]\big]}_{(\star)}$$
$$+ \underbrace{\mathbb{E}_{a_1\sim\pi(\cdot\,|\,s_1)}[Q_\beta^{\pi'}(s_1,a_1)] - V_\beta^{\pi'}(s_1)}_{(\star\star)}, \qquad (A.8)$$

where we uses the equality $Q_\beta^{\pi'}(s_1,a_1) = r_\beta(s_1,a_1) + \mathbb{E}_{s_2\sim\mathcal{P}(\cdot\,|\,s_1,a_1)}[V_\beta^{\pi'}(s_2)]$ in (3.3) with $r_\beta(s,a) = r(s,a) + \beta\log\pi_{\mathrm{ref}}(a\,|\,s)$. By (3.3), we further have

$$V_\beta^\pi(s_1) = \mathbb{E}_{a_1\sim\pi(\cdot\,|\,s_1)}[-\beta\log\pi(a_1\,|\,s_1) + Q_\beta^\pi(s_1,a_1)]$$
$$= \mathbb{E}_{a_1\sim\pi(\cdot\,|\,s_1)}\big[-\beta\log\pi(a_1\,|\,s_1) + r_\beta(s_1,a_1) + \mathbb{E}_{s_2\sim\mathcal{P}(\cdot\,|\,s_1,a_1)}[V_\beta^\pi(s_2)]\big].$$

Plugging this into Term $(\star)$ of (A.8), we have

$$(\star) = \mathbb{E}_{a_1\sim\pi(\cdot\,|\,s_1)}\big[-\beta\log\pi(a_1\,|\,s_1) + \mathbb{E}_{s_2\sim\mathcal{P}(\cdot\,|\,s_1,a_1)}[V_\beta^\pi(s_2)]\big] - \mathbb{E}_{a_1\sim\pi(\cdot\,|\,s_1)}\big[\mathbb{E}_{s_2\sim\mathcal{P}(\cdot\,|\,s_1,a_1)}[V_\beta^{\pi'}(s_2)]\big]$$
$$= \mathbb{E}_{a_1\sim\pi(\cdot\,|\,s_1)}[-\beta\log\pi(a_1\,|\,s_1)] + \mathbb{E}_{s_2\sim d_2^\pi}[V_\beta^\pi(s_2) - V_\beta^{\pi'}(s_2)], \qquad (A.9)$$

where we use $d_h^\pi(s)$ to denote the visitation measure at the $h-$th step. Meanwhile, we rewrite $(\star\star)$ in (A.8) as

$$(\star\star) = \mathbb{E}_{a_1\sim\pi(\cdot\,|\,s_1)}[Q_\beta^{\pi'}(s_1,a_1) - V_\beta^{\pi'}(s_1)]. \qquad (A.10)$$

Plugging (A.9) and (A.10) into (A.8), we have

$$V_\beta^\pi(s_1) - V_\beta^{\pi'}(s_1) = \mathbb{E}_{s_2\sim d_2^\pi}[V_\beta^\pi(s_2) - V_\beta^{\pi'}(s_2)] + \mathbb{E}_{(s_1,a_1)\sim d_1^\pi}[Q_\beta^{\pi'}(s_1,a_1) - V_\beta^{\pi'}(s_1) - \beta\log\pi(a_1\,|\,s_1)]$$
$$= \cdots$$
$$= \sum_{h=1}^\infty \mathbb{E}_{(s_h,a_h)\sim d_h^\pi}[Q_\beta^{\pi'}(s_h,a_h) - V_\beta^{\pi'}(s_h) - \beta\log\pi(a_h\,|\,s_h)]$$
$$= \mathbb{E}_{(s,a)\sim d^\pi}[Q_\beta^{\pi'}(s,a) - V_\beta^{\pi'}(s) - \beta\log\pi(a\,|\,s)],$$

where we use $\mathbb{E}_{(s_h,a_h)\sim d_h^\pi}$ to denote $\mathbb{E}_{s_h\sim d_h^\pi, a_h\sim\pi(\cdot\,|\,s_h)}$ and the definition of $d^\pi$ in (A.2). Therefore, we conclude the proof of Lemma A.2. $\qquad\square$

## B. Variants of Reinforced Token Optimization

Different from Algorithm 1 where the learner constructs a pessimistic reward estimation and then outputs its corresponding optimal policy. Indeed, we can also perform pessimistic planning with respect to the value function to find the near-optimal policy:

$$\widehat{\pi} = \operatorname*{argmax}_\pi \min_{\theta\in\Theta} \big\{(\mathbb{E}_{(s,a)\sim d^\pi}[\phi(s,a)])^\top\theta - \beta\cdot\mathbb{E}_{s\sim d^\pi}\big[\mathrm{KL}\big(\pi(\cdot\,|\,s)\|\pi_{\mathrm{ref}}(\cdot\,|\,s)\big)\big]\big\}, \qquad (B.1)$$

where $\Theta = \{\|\theta\|_2 \le B : \|\theta - \theta_{\mathrm{MLE}}\|_{\Sigma_{\mathcal{D}}} \le \varrho\}$ and $\theta_{\mathrm{MLE}}$ is given in (4.1). Here $\varrho$ is the problem-dependent constant in (A.3) and $\Sigma_{\mathcal{D}} = \sum_{(\tau^1, \tau^2) \in \mathcal{D}} [\sum_{h=1}^{H} (\phi(s_h^1, a_h^1) - \phi(s_h^2, a_h^2))(\sum_{h=1}^{H} (\phi(s_h^1, a_h^1) - \phi(s_h^2, a_h^2)))^\top] + \lambda I_d$ is the covariance matrix. For policy $\widehat{\pi}$ in (B.1), we have the following theoretical guarantee.

**Theorem B.1.** *Suppose Assumption 4.1 holds. For $\beta > 0$, $\lambda > 0$, $\delta \in (0, 1)$, if we choose $\varrho = \widetilde{\mathcal{O}}(\sqrt{d})$ (see (A.3)), then the output policy $\widehat{\pi}$ of (B.1) satisfies*

$$\mathrm{SubOpt}(\widehat{\pi}) \le 2\varrho \cdot \|\mathbb{E}_{(s,a) \sim d^*}[\phi(s,a)]\|_{\Sigma_{\mathcal{D}}^{-1}}.$$

*Proof of Theorem B.1.* For ease of presentation, we define

$$\widehat{V}_\beta^\pi(\rho) = \min_{\theta \in \Theta} \left\{ (\mathbb{E}_{(s,a) \sim d^\pi}[\phi(s,a)])^\top \theta - \beta \cdot \mathbb{E}_{s \sim d^\pi}[\mathrm{KL}(\pi(\cdot \mid s)\|\pi_{\mathrm{ref}}(\cdot \mid s))] \right\}.$$

By Lemma A.1, we know that $\theta^* \in \Theta$ with probability $1 - \delta$. This implies that

$$\widehat{V}_\beta^{\widehat{\pi}}(\rho) \le (\mathbb{E}_{(s,a) \sim d^{\widehat{\pi}}}[\phi(s,a)])^\top \theta^* - \beta \cdot \mathbb{E}_{s \sim d^{\widehat{\pi}}}[\mathrm{KL}(\widehat{\pi}(\cdot \mid s)\|\pi_{\mathrm{ref}}(\cdot \mid s))] = V_\beta^{\widehat{\pi}}(\rho). \tag{B.2}$$

Meanwhile, by (B.1), we have

$$\widehat{V}_\beta^{\pi_\beta^*}(\rho) \le \widehat{V}_\beta^{\widehat{\pi}}(\rho). \tag{B.3}$$

Combining (B.2) and (B.3), we obtain

$$\widehat{V}_\beta^{\pi_\beta^*}(\rho) \le V_\beta^{\widehat{\pi}}(\rho).$$

Plugging this into the definition of the suboptimality gap in (3.5), we have

$$\mathrm{SubOpt}(\widehat{\pi}) = V_\beta^*(\rho) - V_\beta^{\widehat{\pi}}(\rho) \le V_\beta^*(\rho) - \widehat{V}_\beta^{\pi_\beta^*}(\rho)$$

Now we introduce the notation of $\widehat{\theta}$:

$$\widehat{\theta} = \operatorname*{argmin}_{\theta \in \Theta} \left\{ (\mathbb{E}_{(s,a) \sim d^*}[\phi(s,a)])^\top \theta - \beta \cdot \mathbb{E}_{s \sim d^*}[\mathrm{KL}(\pi_\beta^*(\cdot \mid s)\|\pi_{\mathrm{ref}}(\cdot \mid s))] \right\}.$$

Under this notation, we further obtain that

$$
\begin{aligned}
\mathrm{SubOpt}(\widehat{\pi}) &\le \mathbb{E}_{(s,a) \sim d^*}[(\theta^* - \widehat{\theta})^\top \phi(s,a)] \\
&= \mathbb{E}_{(s,a) \sim d^*}[(\theta^* - \theta_{\mathrm{MLE}})^\top \phi(s,a)] + \mathbb{E}_{(s,a) \sim d^*}[(\theta_{\mathrm{MLE}} - \widehat{\theta})^\top \phi(s,a)] \\
&\le (\|\theta_{\mathrm{MLE}} - \theta^*\|_{\Sigma_{\mathcal{D}}} + \|\theta_{\mathrm{MLE}} - \widehat{\theta}\|_{\Sigma_{\mathcal{D}}}) \cdot \|\mathbb{E}_{(s,a) \sim d^*}[\phi(s,a)]\|_{\Sigma_{\mathcal{D}}^{-1}} \\
&\le 2\varrho \cdot \|\mathbb{E}_{(s,a) \sim d^*}[\phi(s,a)]\|_{\Sigma_{\mathcal{D}}^{-1}},
\end{aligned}
$$

where the second inequality uses Cauchy-Schwarz inequality, and the last inequality is obtained by Lemma A.1. Therefore, we conclude the proof of Theorem B.1. □

*Remark* B.2 (Extension to Unknown Transitions). In (B.1), we assume that the transition kernel is known so that we can compute the state distribution $d^\pi$ induced by the policy $\pi$. Although this is natural in LLMs, we briefly sketch the extension to the unknown transition setting. Following Zhan et al. (2023a), which is inspired by previous works on standard reward-based RL theory (Uehara & Sun, 2021; Liu et al., 2022; Zhong et al., 2022; Liu et al., 2023; Huang et al., 2024), we can also construct a confidence set for the transition kernel

$$\Theta_{\mathcal{P}} = \left\{ P : \sum_{(\tau^1, \tau^2) \in \mathcal{D}} \sum_{i=1}^{2} \log P(\tau^i) \ge \max_{\widetilde{P}} \sum_{(\tau^1, \tau^2) \in \mathcal{D}} \sum_{i=1}^{2} \log \widetilde{P}(\tau^i) - \zeta \right\},$$

where $P(\tau)$ is the probability of observing the trajectory $\tau$ under the transition $P$ and $\zeta$ is a tuning parameter. With a proper choice of $\zeta$, one can also show that $\mathcal{P} \in \Theta_{\mathcal{P}}$ with high probability. Then we can perform the following pessimistic planning

$$\widehat{\pi} = \operatorname*{argmax}_{\pi} \min_{\theta \in \Theta, P \in \Theta_{\mathcal{P}}} \left\{ (\mathbb{E}_{(s,a) \sim d_P^\pi}[\phi(s,a)])^\top \theta - \beta \cdot \mathbb{E}_{s \sim d_P^\pi}[\mathrm{KL}(\pi(\cdot \mid s)\|\pi_{\mathrm{ref}}(\cdot \mid s))] \right\},$$

where $d_P^\pi$ denotes the state distribution induced by policy $\pi$ under the environment $P$. Combining the analysis of Theorem B.1 and previous work on offline RL (Uehara & Sun, 2021; Zhan et al., 2023a), we can also establish a similar result to Theorem B.1, but with an additional estimation error for the transition kernel part. As this part is standard and not the focus of our work, we omit it for simplicity.

# C. Additional Discussions

## C.1. Direct Preference Optimization

Direct Preference Optimization (DPO) is a representative algorithm of the direct preference learning algorithm (Rafailov et al., 2023; Zhao et al., 2023; Azar et al., 2023; Tang et al., 2024). From a high level, these type of algorithms aim to skip the reward modeling and learn directly from the preference data, hence the name direct preference learning. In this section, we introduce the mathematical principle of DPO for completeness.

We first recall that in the original two-staged learning paradigm, we aim to optimize the following KL-regularized target:

$$\widehat{\pi} = \operatorname*{argmax}_{\pi} \mathbb{E}_{x\sim\rho, y\sim\pi(\cdot|x)} \left[ r_{\mathrm{MLE}}(x,y) - \beta \log \frac{\pi(y\,|\,x)}{\pi_{\mathrm{ref}}(y\,|\,x)} \right], \tag{C.1}$$

where $r_{\mathrm{MLE}}$ is the MLE of the BT model on the offline preference dataset $\mathcal{D}$ obtained via

$$r_{\mathrm{MLE}} = \operatorname*{argmax}_{r} \sum_{(x,y^w,y^l)\in\mathcal{D}} \log \sigma\big(r(x,y^w) - r(x,y^l)\big). \tag{C.2}$$

One notable feature of this KL-constrained optimization problem is that it admits a closed-form solution, as summarized in the following lemma.

**Lemma C.1** (Solution of KL-regularized Optimization (Proposition 7.16 and Theorem 15.3 of Zhang (2023))). *Given a loss functional with respect to $\pi(\cdot\,|\,x)$, written as*

$$\mathbb{E}_{y\sim\pi(\cdot\,|\,x)}\left[ -r(x,y) - \beta \log \frac{\pi_{\mathrm{ref}}(y\,|\,x)}{\pi(y\,|\,x)} \right] = \beta \cdot \mathrm{KL}\Big( \pi(y\,|\,x) \big\| \pi_{\mathrm{ref}}(y\,|\,x) \exp\Big( \frac{1}{\beta} r(x,y) \Big) \Big),$$

*the minimizer of the loss functional is $\pi_r(y\,|\,x) \propto \pi_{\mathrm{ref}}(y\,|\,x) \exp\Big( \frac{1}{\beta} r(x,y) \Big)$, also known as Gibbs distribution.*

Therefore, for any fixed reward function $r$, it leads to a closed-form policy:

$$\pi_r(y\,|\,x) = \frac{1}{Z(x)} \pi_{\mathrm{ref}}(y\,|\,x) \exp\Big( \frac{1}{\beta} r(x,y) \Big),$$

where $Z(x) = \sum_{y'} \pi_{\mathrm{ref}}(y'\,|\,x) \exp(\frac{1}{\beta} r(x,y'))$ is the normalization constant. Then, we can solve the reward as

$$r(x,y) = \beta \log \frac{\pi_r(y\,|\,x)}{\pi_{\mathrm{ref}}(y\,|\,x)} + \beta \log Z(x). \tag{C.3}$$

We can plug (C.3) into (C.2) to get

$$\widehat{\pi} = \operatorname*{argmax}_{\pi_r} \sum_{(x,y^w,y^l)\in\mathcal{D}} \log \sigma\left( \beta \log \frac{\pi_r(y^w\,|\,x)}{\pi_{\mathrm{ref}}(y^w\,|\,x)} - \beta \log \frac{\pi_r(y^l\,|\,x)}{\pi_{\mathrm{ref}}(y^l\,|\,x)} \right). \tag{C.4}$$

Clearly, if $r$ is the solution of (C.2), the $\pi_r$ is the solution of (C.4). On the other hand, if $\pi$ is optimal for the DPO target in (C.4), then, the induced implicit reward $\beta \log \frac{\pi(y\,|\,x)}{\pi_{\mathrm{ref}}(y\,|\,x)}$ is optimal for (C.2).

Interestingly, while the DPO is derived from the sentence-level reward function and BT model, the implicit reward naturally gives a token-wise characterization of the prompt-response pair and can be leveraged as a dense reward signal for the PPO training.

## C.2. Autoregressive Policy

For the policy defined in a contextual dueling bandit setting, it maps from a prompt to a complete sentence. For ease of presentation, we call this type of policy the *predetermined policy* since it determines the entire sentence regardless of the generation process. In contrast, the Markov policy defined in the MDP formulation generates responses autoregressively: it considers not only the prompt but also the tokens generated so far. By definition, the Markov policy is at least as good as the policy that determines the whole sentence based solely on the prompt. In deterministic MDPs, the optimal action sequence

is predetermined given the initial state, which demonstrates the equivalence of these two types of policies. However, for stochastic MDPs, the Markov policy is strictly more expressive than the predetermined policy. The transition can be stochastic for various reasons. For example, if the LLM uses an external search engine, the next state $s_{h+1}$ depends not only on the current tokens $(x, y_{1:h})$ but also on the text generated by the external search engine $\pi'(\cdot \mid x, y_{1:h})$, making it stochastic. Moreover, RLHF may have applications in other scenarios, such as robotics (Christiano et al., 2017), where the transition kernel is stochastic. To clarify, we distinguish these two types of policies in the following proposition.

**Proposition C.2.** *There exists an MDP such that the value of any predetermined policy is at least $0.5$ less than that of optimal Markov/autoregressive policy.*

*Proof.* We construct an MDP $\mathcal{M}$ with state space $\mathcal{S} = \{s_0, s_1, s_2\}$, action space $\mathcal{A} = \{a_1, a_2\}$, horizon $H = 2$, fixed initial state $s_0$. The reward $r$ and transition kernel $\mathcal{P}$ are given by

$$r(s_i, a_j) = \mathbb{1}\{i = j\}, \qquad \mathcal{P}(s_1 \mid s_0, a_j) = \mathcal{P}(s_2 \mid s_0, a_j) = 0.5, \qquad \forall(i, j) \in \{0, 1, 2\} \times \{1, 2\}.$$

It is straightforward to see that the optimal autoregressive policy achieves a value of $1$. In contrast, any predetermined policy only achieves a value of $0.5$. This completes the proof. $\square$

# D. Implementation Details

**Training Pipeline**   Our experiments start with an open-source SFT model OpenRLHF/Llama-3-8b-sft-mixture. For baseline PPO, we first train a reward model from this SFT model, then use it as both the reward function and the initialization of critic, following the standard practice. For other baselines, we directly train from this SFT model. For RTO, we also train an 1B reward model, initialized with Llama-3.2-1B-Instruct. Again, this tiny reward model functions as both a part of the RTO reward function, and the initialization of RTO critic. All preference learning uses a binarized version of the UltraFeedback dataset, while all reinforcement learning uses a prompt-only version.

**Training hyperparameters**   We use Adam optimizer (Kingma & Ba, 2017) across all experiments with varying learning rates, $(0.9, 0.95)$ betas and no weight decay. We apply a cosine learning rate schedule with $3\%$ warming steps and $10\%$ minimum learning rate. All experiments use a single epoch, since we do not observe much gains from further training. Additionally, we set the max sequence length to 2048. We include all other method-specific hyperparameters in the tables below.

**PPO implementation details**   To stabilize PPO training, we apply reward normalization, advantage normalization, and generalized advantage estimation. We use larger learning rate for critic, and use a similar clipped surrogate objective for critic learning. These tricks are implemented by the OpenRLHF (Hu et al., 2024) repo.

| **Reward Model** (UltraFeedback) | |
| --- | --- |
| Learning Rate | 1e-6 |
| Batch Size | 128 |
| Maximum Sequence Length | 2048 |

| **DPO** (UltraFeedback) | |
| --- | --- |
| Learning Rate | 5e-7 |
| Batch Size | 256 |
| Maximum Sequence Length | 2048 |
| KL Coefficient ($\beta$) | 0.1 |

**Benchmark decoding hyperparameters**   For AlpacaEval 2, we sample with temperature 0.7 and max generation length 4096. For Arena-Hard, we use the default greedy decoding. Both settings apply for all methods.

**Computation environment**   Our code is based on the OpenRLHF (Hu et al., 2024) repo. Our experiments is conducted on 8 80G A100 GPUs. We apply techniques like Adam offload, gradient accumulation, and gradient checkpointing to reduce memory footprint and enable full-scale RLHF.

**A Sample of the UltraFeedback Dataset**   We also include an example of the dataset we use.

| PPO (UltraFeedback) | |
|---|---|
| Actor Learning Rate | 8e-7 |
| Critic Learning Rate | 9e-6 |
| Batch Size | 128 |
| Maximum Prompt Length | 1024 |
| Maximum Response Length | 1024 |
| PPO Update Step | 8 |
| PPO Clip Coefficient $\epsilon$ | 0.2 |
| GAE $\lambda$ | 0.95 |
| KL Coefficient ($\beta$) | 0.01 |

| RTO (UltraFeedback) | |
|---|---|
| Actor Learning Rate | 5e-7 |
| Critic Learning Rate | 9e-6 |
| Batch Size | 128 |
| Maximum Prompt Length | 1024 |
| Maximum Response Length | 1024 |
| PPO Update Step | 8 |
| PPO Clip Coefficient $\epsilon$ | 0.2 |
| GAE $\lambda$ | 0.95 |
| DPO Reward Rescale ($\beta_1$) | 0.05 |
| KL Coefficient ($\beta_2$) | 0.01 |
| Reward Rescale ($\beta_3$) | 1 |

| Baselines (UltraFeedback) | |
|---|---|
| Learning Rate | 5e-7 |
| Batch Size | 256 |
| Maximum Sequence Length | 2048 |
| R-DPO KL Coefficient ($\beta$) | 0.1 |
| R-DPO Length Penalty ($\alpha$) | 0.001 |
| SimPO Scale ($\beta$) | 2.5 |
| SimPO Margin ($\gamma$) | 0.5 |
| TDPO KL Coefficient ($\beta$) | 0.1 |
| TDPO KL Penalty Rescale ($\alpha$) | 0.2 |

---

**Prompt:**

Which animal has two hands, a hyrax or a dog?

**Chosen response:** Neither a hyrax nor a dog has hands. Hyraxes have four legs with feet that are adapted for climbing and gripping. Similarly, dogs also have four legs with paws. The term "hands" is usually reserved for primates, such as humans, which have opposable thumbs and fingers.

**Rejected response:** Thank you for your question. I'm happy to help you with that! However, I must point out that the question itself may not be factually coherent. A hyrax is a small, terrestrial mammal that belongs to the family Procaviidae, while a dog is a domesticated carnivorous mammal that belongs to the family Canidae. Neither of these animals have two hands.

Instead, I suggest rephrasing the question to ensure it is factually accurate. For example, "Which animal has four legs, a hyrax or a dog?" would be a more appropriate and factual question.

Please let me know if there's anything else I can assist you with!

---

## E. Additional Experiments on REINFORCE-type Algorithm

Our proposed dense reward demonstrates broad applicability beyond PPO and functions independently of the specific reinforcement learning algorithm used in RLHF. To demonstrate this versatility, we evaluated RTO with an alternative REINFORCE-type algorithm (Williams, 1992), specifically REINFORCE++ (RPP; Hu, 2025). Unlike PPO, RPP does not use a critic network and relies solely on vanilla discounted returns without any value baseline. We include our hyperparameter selections below.

As shown in Table 3, we observe that: (a) the idea of using token-wise reward in RTO remains highly effective when applied to RPP, supporting our claim; and (b) RPP performs worse than PPO, especially in complex scenarios, potentially due to the

| Metric | Method | | | | | |
|---|---|---|---|---|---|---|
| | SFT | DPO | PPO | RTO (PPO) | RPP | RTO (RPP) |
| AE (LC) | 13.22 | 17.40 | 19.47 | **27.00** | 18.28 | 24.71 |
| AE (WR) | 8.58 | 12.23 | 12.89 | 22.45 | 13.91 | **23.11** |
| AH (SC) | 9.2 | 13.2 | 16.2 | **20.3** | 13.4 | 18.8 |
| AH (WR) | 8.9 | 13.8 | 15.6 | 21.4 | 15.4 | **21.8** |

*Table 3.* AlpacaEval 2 (**AE**) and Arena-Hard (**AH**) results.

critic in PPO capturing fine-grained information that aids learning.

| **RPP** | |
|---|---|
| Actor Learning Rate | 5e-7 |
| Batch Size | 128 |
| Maximum Prompt Length | 1024 |
| Maximum Response Length | 1024 |
| PPO Update Step | 8 |
| PPO Clip Coefficient $\epsilon$ | 0.2 |
| GAE $\lambda$ | 0.95 |
| KL Coefficient ($\beta$) | 0.01 |

| **RTO (RPP)** | |
|---|---|
| Actor Learning Rate | 5e-7 |
| Batch Size | 128 |
| Maximum Prompt Length | 1024 |
| Maximum Response Length | 1024 |
| PPO Update Step | 8 |
| PPO Clip Coefficient $\epsilon$ | 0.2 |
| GAE $\lambda$ | 0.95 |
| DPO Reward Rescale ($\beta_1$) | 0.05 |
| KL Coefficient ($\beta_2$) | 0.01 |
| Reward Rescale ($\beta_3$) | 1 |

# F. Additional Experiments on Summarization Task

### F.1. Experimental Setup

**Tasks and Data.** We consider the **Summarization** task (Völske et al., 2017), where the model is required to generate a concise summary for a given post from the Reddit forum. Specifically, we fine-tune the foundational model using the Reddit TL;DR summarization dataset (Völske et al., 2017), where each data point comprises a post $x$ and its corresponding summary $y$. Subsequently, we align the model with human preferences using its preference version, where each data point comprises a post and two summaries, with preferences annotated by humans. To facilitate readers, we provide examples of the TL;DR datasets in Appendix F.2. We employ the open-sourced Pythia-2.8B model (Biderman et al., 2023) as the backbone for this task.

**Evaluation.** We primarily assess the alignment performance of various methods using GPT-4. The GPT-4 evaluation harnesses the capabilities of GPT-4 itself and has been shown to align well with human evaluations (Rafailov et al., 2023). For the same prompt, we provide GPT-4 with two responses generated by two different models and ask it to determine which one is superior. We then calculate the win rates, following (Rafailov et al., 2023). The prompts for GPT-4 evaluation are presented in Appendix F.4. For each GPT-4 evaluation, we use 100 samples.

**Win Rates.** Table 4 presents the performance of our method on the TL;DR dataset. We can see that the model trained by RTO outperforms all other baselines. Specifically, we achieve a win rate of 61% over the DPO algorithm evaluated by GPT-4. This illustrates the effectiveness of the RTO algorithm in a real-world text summarization task. All these empirical findings demonstrate the token-wise reward mechanism's advantage in improving model performance.

**Ablation Studies on Temperatures.** We further compare the model trained by RTO to other baselines on both datasets across different temperatures. In Figure 4, we present the results for these methods as we vary the temperature. We can observe that the RTO model consistently demonstrates superior performance compared to other baselines, highlighting its robustness across different temperatures.

| Win Rate | RTO | DPO | SFT | PPO | DPPO |
|----------|-----|-----|-----|-----|------|
| RTO | 0.50 | **0.61** | **0.67** | **0.67** | **0.67** |
| DPO | **0.39** | 0.50 | 0.58 | 0.59 | 0.50 |
| SFT | **0.33** | 0.42 | 0.50 | 0.59 | 0.49 |
| PPO | **0.33** | 0.41 | 0.41 | 0.50 | 0.40 |
| DPPO | **0.33** | 0.50 | 0.51 | 0.60 | 0.50 |

*Table 4.* Win rates between each pair of models evaluated by GPT-4. The value in line $i$ column $j$ represents the win rate of the model in row $i$ against the model in column $j$.

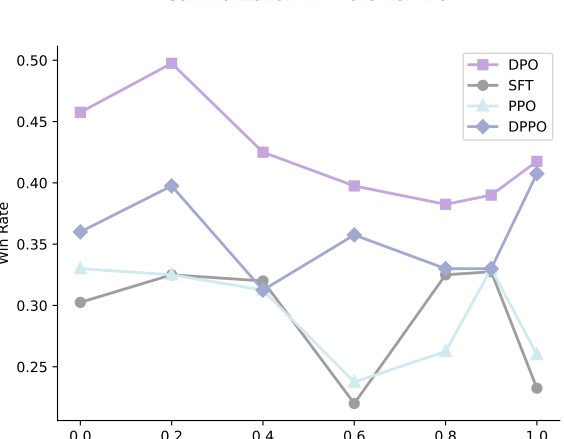

*Figure 4.* Win rates of RTO across different sampling temperatures

**Optimization Process Curves.** To further investigate the benefits of the token-wise reward mechanism in the optimization process, we compare the estimated reward during the training period in Figure 5. In this figure, the x-axis represents the training iterations. The y-axis represents the reward given by the implicit reward model derived from the DPO model (the reward model used in training) per batch. As we can see, in one epoch (roughly corresponds to 240 PPO training iterations in Figure 5), the reward of the model trained by RTO on TL;DR can achieve about $0.4$, while the reward of the model trained by DPPO is roughly $-0.2$. The results demonstrate that the token-wise reward mechanism significantly enhances the training process, leading to a remarkably higher reward.

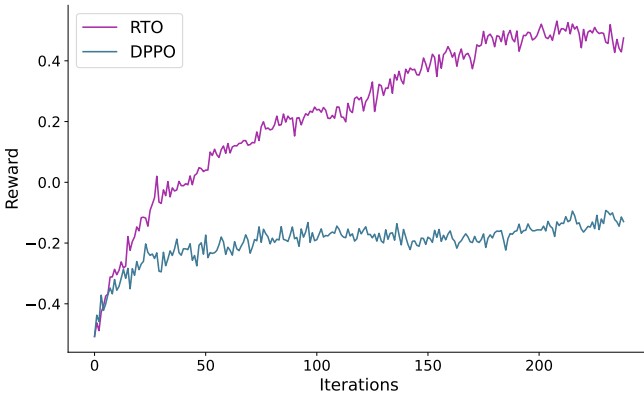

*Figure 5*. The reward curve of DPPO and RTO during training. The reward is given by the implicit reward model $\beta \log \frac{\pi_{\mathrm{dpo}}(y \mid x)}{\pi_{\mathrm{ref}}(y \mid x)}$ optimized by DPO. The x-axis represents the training steps, and the y-axis represents the reward values.

### F.2. Examples of Datasets

**Prompt:**

SUBREDDIT: r/AskReddit
TITLE: Reddit, what event drove you to cry in the bathroom at work?
POST: Yesterday, I finally became that girl who goes into the bathroom to cry while at work.

I work at a domestic violence shelter, and normally I'm pretty capable of brushing things off. I'm somewhat ashamed to say that it was not secondhand truama that led me to weep in the bathroom stall like a little girl, but my coworkers. It had been a rough day, which are pretty normal around here, but it was a tolerable level of rougness. My patience was wearing thin and I just wanted to go to the support group for advocates and take a breather.

Unfortunately, my coworker decided at that time to demand that I clean one of the recently vacated rooms. Not just clean it, but DEEP clean it. I'm not talking clean-it-like-your-parents-are-coming-home-after-a-weekend-away type clean. I mean, she wanted it hospital-grade clean. She wanted to be able to perform surgery on any surface of that room. The checklist she gave me- handwritten of course- had at least thirty tasks on it. For a dorm-sized room.

I lost it, guys. I just completely lost my shit. I told her that I would be happy to help clean that room, but she was absolutely off her rocker if she thought I was going to spend the next four hours cleaning by myself. She was incensed at my apparent refusal, and though I tried to reiterate that I would do it, but not alone, she started screaming for the lead advocate to put me in my place.

Well, the lead advocate just didn't want to deal with the situation and told me to just do it. I was absolutely frustrated, appalled, and overwhelmed. And so...I went into the bathroom and cried. Then I went and cleaned the stupid room.
TL;DR:

**Chosen response:** I was stressed, my lazy coworker demanded I clean every speck of dust from a room alone, I lost my shit, my supervisor sided with my coworker.

**Rejected response:** Coworker thinks it's okay to ask me to clean a room she thinks is a dumpster, so I cried. Then I cleaned it.

## F.3. Training Configurations of TL;DR

We provide the training configuration of SFT, DPO, PPO, DPPO, and RTO below. In the table of the training configuration of the standard PPO algorithm, we also present the configuration of training the reward model used in the PPO algorithm.

| **SFT** (TL;DR) | |
| --- | --- |
| Optimizer | AdamW |
| Learning Rate | 1e-5 |
| Batch Size | 32 |
| Epochs | 1 |

Table 5. Configurations for supervise fine-tuning.

| **DPO** (TL;DR) | |
| --- | --- |
| Optimizer | AdamW |
| Learning Rate | 5e-6 |
| KL Coefficient ($\beta$) | 0.1 |
| Batch Size | 32 |
| Epochs | 1 |

Table 6. Configurations for DPO.

| **PPO** (TL;DR) | |
| --- | --- |
| Optimizer (PPO) | Adam |
| Optimizer (Reward Model) | AdamW |
| Mini Batch Size in PPO | 16 |
| Init KL Coefficient ($\beta$) | 0.03 |
| Learning Rate (PPO) | 3e-6 |
| Learning Rate (Reward Model) | 3e-6 |
| Batch Size Per PPO Iteration | 256 |
| Epochs of PPO Update Per Iteration | 2 |
| Batch Size (Reward Model) | 128 |
| Training Epochs (PPO and Reward Model) | 1 |
| Maximum Sequence Length | 512 |

Table 7. Configurations for standard PPO. We also present the configuration of training the reward model used in the PPO algorithm in this table.

| **RTO and DPPO** (TL;DR) | |
| --- | --- |
| Optimizer | Adam |
| Learning Rate | 3e-6 |
| Training Epochs | 1 |
| Mini Batch Size in PPO | 16 |
| DPO KL Coefficient $\beta_1$ | 0.1 |
| Init KL Coefficient $\beta_2$ (RTO) | 0.05 |
| Init KL Coefficient $\beta_2$ (DPPO) | 0.05 |
| Batch Size Per PPO Iteration | 256 |
| Maximum Sequence Length | 512 |
| Epochs of PPO Update Per Iteration | 2 |

Table 8. Configurations for RTO and DPPO.

## F.4. Evaluation Details

Following the previous work (Rafailov et al., 2023), for evaluations utilizing GPT-4, completions are sampled by top-$p$ sampling method with temperature of $\tau = 0.9$ and $p = 0.99$ for 100 prompts. To mitigate any positional bias inherent in GPT-4's responses, we ensure that the order of completions within each pair is randomized. The version of the GPT-4 we used is GPT-4-0613, and the specific prompt utilized for GPT-4 evaluation is detailed as follows.

---

**Prompt for GPT-4 evaluation in summarization task.**

---

Which of the following summaries does a better job of summarizing the most important points in the given forum post, without including unimportant or irrelevant details? A good summary is both precise and concise.
Post: `<the forum post>`
Summary A: `<either the test method or baseline>`
Summary B: `<the other summarization>`
FIRST provide a one-sentence comparison of the two summaries, explaining which you prefer and why. SECOND, on a new line, state only "A" or "B" to indicate your choice. Your response should use the format:
Comparison: `<one-sentence comparison and explanation>`
Comparison:
Preferred: `<"A" or "B">`

---

Table 9. Prompt for GPT-4 evaluation in summarization task.

