# OpenReview forum: "DPO Meets PPO: Reinforced Token Optimization for RLHF"
_ICML.cc/2025/Conference — ICML 2025 spotlightposter_

### Official Review · Reviewer_E8vv · 2025-03-08

**Overall Recommendation:** 3

**Summary:**

This paper develops an RLHF framework with a fine-grained token-wise reward characterization. Specifically, they model RLHF as an MDP, offering a more precise token-wise characterization of the LLM’s generation process. They introduce RTO algorithm, which extracts token-wise reward signals from offline preference data and subsequently performs RL training with respect to the learned token-wise rewards. In practice, they will introduce a practical implementation of RTO, which uses a token-wise reward extraction approach from DPO.

## update after rebuttal: Thanks for the rebuttal. I do not have any further question. I will keep my original rating.

**Claims And Evidence:**

Yes

**Essential References Not Discussed:**

See my comments in "Questions For Authors".

**Experimental Designs Or Analyses:**

The experiment looks good.

**Methods And Evaluation Criteria:**

Yes.

**Other Comments Or Suggestions:**

There are typos in Section 5. One of the competitive methods is called token-wise DPO (TDPO). The authors used "DDPO" and "DPPO" in plots and discussions. Are they all different?

**Other Strengths And Weaknesses:**

See my comments in "Questions For Authors".

**Questions For Authors:**

1. Assumption 3.1 is not a rigorous assumption. The parameters $A$ and $\xi$ have not been defined. What are their possible ranges of values? The current statement is not an assumption without conditions on $A$ and $\xi$. In addition, it is helpful to add more discussions on the intuition of this assumption and why it makes sense in the considered problem.

2. There is a gap between the Algorithm 1 (Theoretical Version) and Algorithm 2 (Practical Version). The theorems were proved for Algorithm 1, whole all experiments were done based on Algorithm 2. If Algorithm 1 is not implementable in practice, why to introduce it? Can you prove the suboptimal gap bound for the Algorithm 2 directly?

3. A few highly related key references were only mentioned in Section A of the Appendix. For example, Zeng et al. (2024) also considered token-level DPO. Can authors add more discussions on the difference and the technical novelty beyond this paper?

**Relation To Broader Scientific Literature:**

See my comments in "Questions For Authors".

**Theoretical Claims:**

See my comments in "Questions For Authors".

---

> ### Author Rebuttal · Authors · 2025-04-01
>
> Thank you for your review and support. Below are our response to your questions.
>
> **Q1:** Assumption 3.1 is not a rigorous assumption. The parameters $A$ and
> $\xi$ have not been defined. What are their possible ranges of values? The current statement is not an assumption without conditions $A$ and $\xi$. In addition, it is helpful to add more discussions on the intuition of this assumption and why it makes sense in the considered problem.
>
> **A1:** The parameter $A = |\mathcal{A}|$ is the action set size (Line 220), and $\xi$ is a constant introduced in Assumption, to clarifiy this, we revise Assumption 3.1 as
>
> Assumption 3.1. There exists a response $y = y_{1:H}$ satisfying $\pi^*(y|x) \ge A^{-\xi}$ for some $0 \le \xi \le H$.
>
> We have also stated in the paper that "By the pigeon-hole principle, there must be a response $y$ such that $\pi^*(y | x) \ge A^{−H}$, implying that $\xi ≤ H$ naturally. In practice, $\xi$ is usually much smaller than $H$ because the language model tends to choose the optimal response rather than making a random guess."
>
> More specifically, there are at most $A^H$ possible responses/trajectories $y = y_{1:H}$ (each step has $A$ actions (tokens) to choose from and the sequence length is at most $H$). Therefore, at least one response is chosen with a probability of at least $A^{-H}$. Moreover, given a prompt, the LLM will only generate several likely tokens with high probability, which makes the final generation probability of the most likely response significantly larger than the random guess probability $A^{-H}$. We denote this as $A^{-\xi}$ with $\xi \ll H$.
>
>
>
> **Q2:** There is a gap between the Algorithm 1 (Theoretical Version) and Algorithm 2 (Practical Version). The theorems were proved for Algorithm 1, whole all experiments were done based on Algorithm 2. If Algorithm 1 is not implementable in practice, why to introduce it? Can you prove the suboptimal gap bound for the Algorithm 2 directly?
>
> **A2:** We introduce the practical version of the theorem primarily to demonstrate that *token-level MDPs with KL constraints can be learned sample-efficiently*. We believe this constitutes a new and important theoretical advancement that deserves emphasis. We also acknowledge that proving the suboptimality gap bound for Algorithm 2 directly is challenging; in fact, even for standard PPO, its theoretical analysis often requires simplifications, such as replacing the GAE estimation with an optimistic estimation. Thank you for your question, and we will highlight our motivation for introducing the theoretical version and discuss its current limitations.
>
>
> **Q3:** A few highly related key references were only mentioned in Section A of the Appendix. For example, Zeng et al. (2024) also considered token-level DPO. Can authors add more discussions on the difference and the technical novelty beyond this paper?
>
> **A3:** Sure. We will include more discussion in the revision. Meng et al. (2024) propose SimPO by modifying the DPO objective, replacing the reference model with response length, and adding a margin threshold. Zeng et al. (2024) also consider a token-level reward and leverage this insight to develop token-level DPO, which performs better than the original DPO. These are beyond the scope of our work. In contrast, we utilize the implicit token-level reward provided by the original DPO as the dense token-level reward for RL training.
>
> **Q4:** The authors used "DDPO" and "DPPO" in plots and discussions. Are they all different?
>
> **A4:** Thanks for pointing out the typo and we will correct them. Both "DDPO" and "DPPO" indicate the baseline that uses RTO reward delayed to the last token.

---

### Official Review · Reviewer_J8vS · 2025-03-10

**Overall Recommendation:** 4

**Summary:**

Summary: The authors propose a token-level MDP formulation for LLM. post-training. They use the token-level action probabilities from a Direct Preference Optimization (DPO) trained LLM. Authors argue that the current formulation of LLM post-training is closer to a contextual bandit than it is to a reinforcement learning, and hence does not utilize the full power of the RL machinery, i.e. credit assignment, advantage function etc. Further, authors propose a practical implementation of their method: RTO, where practitioners can directly plug-in DPO token-level probability estimates in a standard PPO implementation.

## update after rebuttal

All reviewers seem to agree that this is an interesting paper for RL fine-tuning of LLM. My initial assessment of accept has not changed.

**Claims And Evidence:**

The claims made by the authors, i.e. a token-level reward is better than sentence-level reward for LLM post-training is well supported by the experimental claims. Authors compare with both “RL-free” methods like (DPO, SimPO), and the standard online RL-method (PPO).

**Essential References Not Discussed:**

NA

**Experimental Designs Or Analyses:**

The experimental design and the analysis of the proposed method RTO, including the sample efficiency, reward granularity are sound.

**Methods And Evaluation Criteria:**

The proposed method is a very sensible extension of the existing sentence-level PPO fine-tuning, and the experimental setup is sound.

**Other Comments Or Suggestions:**

NA

**Other Strengths And Weaknesses:**

1. The paper is very well-written, especially the introduction and discussion of preliminary work. In my experience, most papers do not explicitly spell out that the most of the current setup for LLM post-training is in fact a contextual bandit setup, and not a MDP setup, so it's nice to see this assumption explicitly spelled out.

2. In section 3.3., authors prove that the sentence-level MDP formulation is sample inefficient in terms of sample complexity, and a token-level formulation is required.

3. The paper has a nice mix of both solid theoretical analysis, and practical algorithm for implementation. In my experience, some papers (on this topic) are either purely empirical, or purely theoretical. Nothing wrong with it, but it's nice to see a paper with a good mix of both.

**Questions For Authors:**

1. I have a question on the results. SimPO method outperforms PPO by a large margin (Table 1). Do you have any intuition for this? Literature seem to suggest that RL-free offline methods are outperformed by online methods, such as PPO, but the results in this paper seem to suggest differently.

2. Is Assumption 4.1 valid for only linear reward models, or does it also apply for LLM-based reward model (with \pi fixed, and \theta parameter)?

**Relation To Broader Scientific Literature:**

The paper improves the post-training method for LLMs, and given the significance of LLMs recently, the presented method can have a broad impact.

**Theoretical Claims:**

I have checked the theoretical claims: Proposition 3.2 (sentence-level post-training has a much higher sample complexity than token-level post-training), Theorem 4.2 (Sub-optimality of token-level rewards), Eq. 4.7 (practical version of token-level rewards), and they all make sense to me.

---

> ### Author Rebuttal · Authors · 2025-04-01
>
> Thank you for your review and support. Below are our response to your questions.
>
> **Q1:** I have a question on the results. SimPO method outperforms PPO by a large margin (Table 1). Do you have any intuition for this?
>
> **A1:** Our observation is that the RL-free methods are good at fitting the **style** over the deep RL methods, while the human/GPT-4 preference are easily hacked by the preference bias (see Figure 3 of [1] and related discussions). Compared to the DPO, SimPO further removes the KL constraint so it is more easily to achieve a higher in-domain score (AlpacaEval). However, we also notice that SimPO will also often hurt the OOD reasoning performance (see Table 9 of their paper for example).
>
> [1] From Lists to Emojis: How Format Bias Affects Model Alignment
>
> **Q2:** Literature seem to suggest that RL-free offline methods are outperformed by online methods, such as PPO, but the results in this paper seem to suggest differently.
>
> **A2:** Indeed, almost all closed-source state-of-the-art LLMs, including ChatGPT, GPT-4, and the recent DeepSeek R1, are trained using RL-based methods like PPO and GRPO. However, the open-source community has struggled to replicate these RL training techniques effectively. Our work aims to narrow this gap by exploring both dense and sparse reward approaches.

---

### Official Review · Reviewer_wuwo · 2025-03-13

**Overall Recommendation:** 3

**Summary:**

The paper introduces Reinforced Token Optimization (RTO), a framework that integrates Direct Preference Optimization (DPO) and Proximal Policy Optimization (PPO) to improve Reinforcement Learning from Human Feedback (RLHF). The authors argue that existing RLHF implementations using PPO underperform due to a mismatch between sentence-level reward modeling (bandit formulation) and PPO’s requirement for token-wise rewards. RTO addresses this by reformulating RLHF as a Markov Decision Process (MDP) with token-level rewards. Key contributions include:

- MDP formulation
- RTO algorithm
- Theoretical guarantees
- Empirical results

**Claims And Evidence:**

The claims are supported by theoretical analysis and empirical validation:

- MDP superiority: Theoretically justified via comparisons between bandit and MDP formulations (Section 3.3).
- RTO’s effectiveness: Experiments on standard benchmarks (AlpacaEval 2, Arena-Hard) and ablation studies (e.g., data scaling) validate performance gains.
- Token-wise reward extraction: The use of DPO to derive token-level rewards (Appendix D.1) is plausible but requires deeper scrutiny (see Q4/Q5).

**Essential References Not Discussed:**

None

**Experimental Designs Or Analyses:**

- Baselines: Comparisons with PPO, DPO, R-DPO, and SimPO are reasonable, but implementation details (e.g., reward models, KL penalties) are sparse.
- Data efficiency: The claim that RTO achieves PPO-level performance with 1/8 of the data is compelling but requires validation across multiple seeds.
- Ablation studies: Limited analysis of RTO’s components (e.g., DPO’s role in token-wise rewards).

**Methods And Evaluation Criteria:**

- Methods: RTO’s integration of DPO and PPO is novel and addresses the token-wise reward gap in RLHF. The MDP formulation aligns with LLMs’ autoregressive nature.
- Evaluation: Benchmarks (AlpacaEval 2, Arena-Hard) are standard for LLM alignment. However, details on baseline implementations (e.g., PPO hyperparameters) and statistical significance are unclear.

**Other Comments Or Suggestions:**

None

**Other Strengths And Weaknesses:**

Strengths:

- Novel integration of DPO and PPO;
- strong empirical results;
- addressing a critical RLHF limitation.

Weaknesses:
- Theoretical gaps;
- limited ablation studies;
- insufficient implementation details for reproducibility.

**Questions For Authors:**

1. Theoretical Proofs: Could you provide a complete proof for the sample efficiency claim in Appendix B?

2. DPO’s Role: How does DPO-derived token-wise reward correlate with human preferences? Is there empirical validation beyond benchmark scores?

3. Baseline Details: Were PPO and DPO baselines trained with identical hyperparameters (e.g., KL penalty, reward scaling)?

Responses to these questions could strengthen the paper’s theoretical grounding and empirical rigor.

**Relation To Broader Scientific Literature:**

The work builds on RLHF (Ziegler et al., 2019), PPO (Schulman et al., 2017), and DPO (Rafailov et al., 2023). It advances the field by addressing the token-level reward sparsity problem, a known limitation of PPO in RLHF.

**Theoretical Claims:**

The paper claims RTO’s sample efficiency (Section 4) but lacks formal proofs. The theoretical analysis in Section 3.3 (MDP vs. bandit) is intuitive but not rigorously proven. Appendix B mentions a "near-optimal policy" guarantee but does not provide a full proof.

---

> ### Author Rebuttal · Authors · 2025-04-01
>
> Thank you for reviewing our paper. Below are our responses.
>
> **Question Regarding Theory:** Theoretical Proofs: Could you provide a complete proof for the sample efficiency claim in Appendix B?
>
> **Response:** We have provided complete and rigorous proofs for both Proposition 3.2 and Theorem 4.2 in Appendix B. Should you have any specific questions or require further clarification, we are happy to address them. Regarding our extension mentioned in Remark B.3, we have acknowledged that this is a standard result in RL literature (see e.g., Theorem 2 in [1] for a detailed proof). Given that this standard extension is not central to the main claims of our paper, we opted to provide a concrete reference rather than including a detailed proof. We appreciate your understanding.
>
> [1] Fast Global Convergence of Natural Policy Gradient Methods
> with Entropy Regularization. https://arxiv.org/pdf/2007.06558
>
>
> **Question Regarding Experiments:**(1) DPO’s Role: How does DPO-derived token-wise reward correlate with human preferences? Is there empirical validation beyond benchmark scores? (2) Baseline Details: Were PPO and DPO baselines trained with identical hyperparameters (e.g., KL penalty, reward scaling)? (3) Data efficiency: The claim that RTO achieves PPO-level performance with 1/8 of the data is compelling but requires validation across multiple seeds.
>
> **Response:** Thank you for your question.
>
> (1): We have added the following results to illustrate that DPO-derived token-wise rewards effectively capture human preferences. For the pairwise data, we measure consistency between human choices and DPO reward choices (reward implies an order). We observed a 79.23% consistency in the training data and 72.72% in the test data. Additionally, we want to emphasize that the performance of DPO-derived token-wise rewards used in reinforcement learning training, evaluated by widely recognized benchmarks, is a key standard for assessing its quality and is also the central focus of our work. To further illustrate the generalizability of DPO-derived reward, we've extended our methods to another RL algorithm REINFORCE++ [1].
>
> The following table presents the Alpaca Eval 2 benchmark scores, further demonstrating the ability of DPO rewards to effectively capture human preferences and the broader applicability of our RTO method.
>
> | Algorithm | AE2 LC | AE2 WR |
> | -------- | -------- | -------- |
> | REINFORCE++ | 18.28 | 13.91 |
> | RTO (with REINFORCE++ objective) | 24.71 | 23.11 |
>
> [1] REINFORCE++: A Simple and Efficient Approach for Aligning Large Language Models
>
>
> (2): We have included the detailed hyperparameter choices in Appendix E. The DPO and PPO hyperparameters differ, and both sets were derived from well-tuned hyperparameters provided by the OpenRLHF project. For your convenience, we present important hyperparameters here: The KL penalty coefficient ($\beta$) is $0.01$ for PPO and $0.1$ for DPO. Other PPO hyperparameters are clip coefficient $\epsilon=0.2$ and GAE $\lambda=0.95$. We train a reward model for PPO, and another tiny reward model for RTO ($r_\text{MLE}$) following the standard practice. The reward scaling of PPO is $1$ for reward model and $\beta=0.01$ for KL reward penalty. For RTO, we use $\beta_3=1$ for reward model, $\beta_2=0.01$ for KL reward penalty, and $\beta_1=0.05$ for DPO reward.
>
>
> (3): Thanks for the suggestions. Our implementation is based on OpenRLHF and our (also the community's) experience (across hundrends of LLM RL training in either preference learning or math/coding task) is that with the well-tuned recipe, the LLM RL is very stable and not greatly affected by the choice of random seed. As a result, we believe our findings are sufficient to demonstrate the sample efficiency of our claim.

---

### Official Review · Reviewer_M1p9 · 2025-03-16

**Overall Recommendation:** 3

**Summary:**

The paper presents RTO, a novel reward learning method for tuning LLMs from preference data. Such a process involves two stages: reward modeling and an RL step (typically PPO).  The main novelty of this paper lies in modifying the Bradley-Terry (BT) loss in the reward modeling step to yield a reshaped token-level reward instead of a single per-example reward. This allows the user authors to establish a more direct connection to the second RL step. This results in theoretical results for the regret under the linear reward assumptions. A nice feature of their theoretical approach is that it explicitly considers the KL-regularized version of RL used for LLM fine-tuning, but not so common in general RL.

**Claims And Evidence:**

### Claims

The authors main claims are that (1) they propose a framework for RLHF as an MDP; (2) demonstrate near optimal sample efficiency under this model; (3) achieve improvements in well-established benchmarks under the new reward model.


### Evidence

**Regarding (1)**

I think the contribution should be made more precise and I found it very confusing at first read.

 It is true that the reward modeling step assumes a bandit structure, but this step is only for the reward learning stage, and the bandit structure itself is quite irrelevant. The RL (PPO) step in current RLHF is already doing RL over a token-level MDP by assigning a reward per token (with zero reward but the last token).

I also question that the current proposed reward modeling step is framed as an MDP. This is so because the MLE objective in (4.1) is still at the trajectory level. There is simply no way to mathematically define it for a partial trajectory. If it was truly at the transition level, I could apply it over a random sample of transitions. But their framework still requires to randomly sample over trajectoreis. Could the authors comment ton that?

As another comment, the authors themselves state that:

> ““There have also been previous works (Pacchiano et al., 2021; Chen et al., 2022; Wang et al., 2023; Li et al., 2023c; Zhan et al., 2023a) studying RLHF under the MDP framework, also known as dueling RL and preference-based RL. However, these works do not consider the KL constraint, which is an essential component of RLHF.”

Hence, the initial claim in (1) should be made consistent with this statement.
Ultimately, I think this paper should be framed more simply as a a reward modeling strategy for RLHF or a reward shaping strategy.

**Regarding (2)**

* I think this statement is slightly imprecise since the optimality bound of Theorem 4.2 depends on a tuning parameter lambda which is not part of the reward model r.

* I also don’t understand very well the value of Assumption 4.1 (Linear reward) since the authors proposing exactly how to fit the reward model under the proposed modified BT/DPO at token-level. Shouldn’t the assumption be true or false by construction?

**Regarding (3)**

The experiments look promising, but they are also quite limited since they only apply to baseline Llama 8B and a relative small number of baselines. See my comment on the experiments section.

**Essential References Not Discussed:**

As far as I am aware, the paper conducts a comprehensive review.

**Experimental Designs Or Analyses:**

Some additional experiments could improve the practical value of the method:

1.	Experiment with additional base LLMs. Relying solely on Llama 8b is very limiting. Ideally there would more baseline models and a larger LLM.
2.	Does the performance gains also hold if an alternative RL method for step 2 is applied (e.g. RLOO [1]). PPO is probably not the state of the art.
3.	How sensitive is the performance to the hyperparameter $\beta_3$?


[1] Ahmadian, et al. (2024). Back to basics: Revisiting reinforce style optimization for learning from human feedback in llms. arXiv preprint arXiv:2402.14740, 2024.

**Methods And Evaluation Criteria:**

Yes

**Other Comments Or Suggestions:**

*Minor suggestion*:  I was a bit thrown off by the phrase “After examining the open-source implementation of PPO” in the introduction. I eventually understood the intention of the authors. But first of all, I don’t know what open source has to do with it. Second, PPO is not an ambiguous algorithm; in fact, the authors are implementing it in their paper without modification. The real issue is the mismatch between the “sentence-level reward modeling stage” and the required token-level reward in PPO, which results in the potentially inefficient reward assignment to the last token only.

**Other Strengths And Weaknesses:**

N/A

**Questions For Authors:**

I have already included my questions in the other sections.

Additional questions:

1.	How does the role of $\lambda$ hyperparameter in the theoretical version carries to the practical version?
2.	If the main value of the framework is the reward shaping, could I obtain a similar performance by applying reward shaping techniques (e.g., potential based) to the sentence-level reward?

**Relation To Broader Scientific Literature:**

The paper is very timely given the centrality of LLM finetuning in modern AI methods across various fields.

**Theoretical Claims:**

I skimmed over the theoretical derivation, but did not check them carefully.

Note that $\xi$ in Assumption 3.1 is never defined. What is it? Can you define it formally and provide an intuition since it is so import to the claim of sample complexity optimality?

---

> ### Author Rebuttal · Authors · 2025-04-01
>
> Thank you for reviewing our paper. We summarize your questions and address them as follows.
>
> **Q1:** MDP fomulation and Previous Theoretical Work
>
> **A1:**  We agree with the reviewer that the existing PPO is implemented in an *MDP with zero reward for all but the last token*, but we would like to claim that this is *essentially a bandit problem*. This point is explicitly stated in Section 3.5 of the seminal InstructGPT paper [1]. In contrast, a token-level MDP with dense rewards is a real multi-step decision-making problem (see Proposition 3.2 for a theoretical gap).
>
> We will emphasize the importance of KL constraint in claim (1), to different from existing theoretical work on dueling RL. Moreover, we would like to emphasize that these works are purely theoretical works and does not lead to corresponding paractical algorithm leveraging dense rewards.
>
> Finally, the token-level MDP terminology is mainly used to emphasize the use of dense rewards, in contrast to the bandit formulation with sparse rewards, which aligns with the reviewer's understanding. This token-level MDP was rigorously introduced first in our paper, and it is well recognized by the community [2]. Therefore, we prefer to maintain the terminology of token-level MDPs with KL constraints while highlighting the differences from existing bandit formulations and theoretical dueling RL work.
>
> [1] Training language models to follow instructions with human feedback
>
> [2] Value-incentivized preference optimization: A unified approach to online and offline rlhf
>
> **Q2:** MLE in (4.1) at the trajectory level
>
> **A2:** The MLE objective in (4.1) looks like at the trajectory level because the preference signal is usually given in the trajectory level and the cumulative token-level reward is the trajectory-level reward. However, even the trajectory-based preference reward can naturally induce a cumulative token-level reward. This is more evident for problems with step-wise structure like mathematical reasoning [3].
>
> [3] Entropy-Regularized Process Reward Model
>
> **Q3:** Theorem 4.2 depends on $\lambda$ and the role of $\lambda$
>
> **A3:** The parameter $\lambda$ is just a regularization parameter for theory (ensure $\Sigma_{\mathcal{D}}$ has inverse), and our Theorem 4.2 holds for any $\lambda>0$. You can simply regard $\lambda = 1$.
>
> **Q4:** Verify Assumption 4.1 by construction
>
> **A4:** If we choose $\phi(s, a) = 1_{(s, a)}$ as the one-hot vector and $\theta_{(s, a)}=r(s, a)$,  then the assumption is satisfied with $d = |\mathcal{S}| \times |\mathcal{A}|$. However, the reward function may exhibits some low-dimensional respresentation $\phi \in \mathbb{R}^d$ with $d < |\mathcal{S}| \times |\mathcal{A}|$, and thus we assume Assumption 4.1 with potentially small $d$ and do not make the explicit construction. This is also the main motivation of linear bandit and linear MDP.
>
> **Q5:** Experiments: (1) limited base model and baselines. (2) application to other RL algorithms; (3) hyperparameter $\beta_3$?
>
> **A5:** (1) Regarding baseline selection, we focused on the most relevant (DPO, PPO), strong (R-DPO, SimPO), and concurrent (SimPO, TDPO) methods. Given the limited performance of numerous other alignment algorithms reported in SimPO, we opted not to include them in our comparisons. Due to time limit, we are still trying to add experiments on other base models. We appreciate your understanding.
> (2) We extend our experiments to REINFORCE++ (RF++) [5], an alternative RL algorithm. We selected RF++ since
> - Unlike reasoning tasks, chat alignment typically does not involve sampling multiple responses per prompt like RLOO and GRPO.
> - RF++ exhibits significant differences from PPO, notably the absence of a critic model.
>
> The table below shows the power of RTO when applied to RF++.
>
> | Algorithm | AE2 LC | AE2 WR |
> | -------- | -------- | -------- |
> | RF++ | 18.28 | 13.91 |
> | RTO | 24.71 | 23.11 |
>
> [4] REINFORCE++: A Simple and Efficient Approach for Aligning Large Language Models
>
> (3)  Since only relative ratio matters and to minimize tuning, we fix $\beta_2$ the same value PPO uses, choose $\beta_3 = 1$, and tune $\beta_1$ (see discussion below (4.7)). In our internal experiments we found our algorithm are quite robust to the choice of $\beta_1$. Thus we believe that conversely, RTO is not sensitive to the choice of $\beta_3$.
>
> **Q6:** Reward shaping
>
> **A6:** Our algorithm can be interpreted as a form of potential-based reward shaping, with $F(s, s' = (s, a)) = \Phi(s') - \Phi(s) = \log \frac{\pi(s')}{\pi_{\mathrm{ref}}(s')}-\log \frac{\pi(s)}{\pi_{\mathrm{ref}}(s)}=\log\frac{\pi(a|s)}{\pi_{\mathrm{ref}}(a|s)}$, where $\Phi$ is the potential function and $F(s, s' = (s, a))$ denotes the token-wise reward function added to each token. Thank you for the insightful question and we will emphasize this in the revision.
>
> Regarding your minor suggestion, we would emphasize the issue of the reward modeling type directly, rather than mentioning the open-source implementation.

---

> > ### Comment · Reviewer_M1p9 · 2025-04-03
> >
> > Thank you for your response. My questions have been resolved so I have increased my score.

---

> > > ### Author Response · Authors · 2025-04-03
> > >
> > > Thank you for reading our response. We truly appreciate your support and are glad we could address your questions.

---

### Decision · Program_Chairs · 2025-05-01

**Decision:**

Accept (spotlight poster)

**Comment:**

This paper aims to address the mismatch between sentence-level reward modeling in RLHF and the use of PPO by reformulating RLHF as an MDP with token-level rewards. The reviewers unanimously appreciated the paper for its clarity, the novelty of the proposed approach, and the significance of its contribution. However, they suggested that the authors clarify the precise contribution of the paper, particularly in distinguishing their method from the PPO step used in current RLHF pipelines and from related work in dueling RL. The authors are also encouraged to incorporate the reviewers’ comments on the experiments in their revised version.